# iLOCO: Distribution-Free Inference for Feature Interactions

## Abstract

Feature importance measures are widely studied and are essential for understanding model behavior, guiding feature selection, and enhancing interpretability. However, many machine learning fitted models involve complex interactions between features. Existing feature importance metrics fail to capture these pairwise or higher-order effects, while existing interaction metrics often suffer from limited applicability or excessive computation; no methods exist to conduct statistical inference for feature interactions. To bridge this gap, we first propose a new model-agnostic metric, interaction Leave-One-Covariate-Out (iLOCO), for measuring the importance of pairwise feature interactions, with extensions to higher-order interactions. Next, we leverage recent advances in LOCO inference to develop confidence intervals for our iLOCO metric, which is asymptotically valid without making strong assumptions on the data generating model or the implemented machine learning algorithm. To address computational challenges, we also introduce an ensemble learning method for calculating the iLOCO metric and confidence intervals that we show is both computationally and statistically efficient. We validate our iLOCO metric and our confidence intervals on both synthetic and real data sets, showing that our approach outperforms existing methods and provides, to the best of our knowledge, the first inferential framework with formal validity guarantees for detecting model-agnostic feature interactions.

## 1 Introduction

Machine learning systems are increasingly deployed in critical applications, necessitating human-understandable insights. Consequently, interpretable machine learning has become a rapidly expanding field (Murdoch et al., 2019; Du et al., 2019). For predictive tasks, one of the most important aspects of interpretation is assessing feature importance. While much focus has been placed on quantifying the effects of individual features, the real power of machine learning lies in its ability to model higher-order interactions and complex feature representations (Grömping, 2009). However, there is a notable gap in interpretability: few methods provide insights into the higher-order feature interactions that truly drive model performance.

Understanding how features interact to influence predictions is a critical component of interpretable machine learning, particularly in scientific applications or to make downstream business decisions. In drug discovery, for example, uncovering how chemical properties interact can guide the identification of effective compounds in large-scale screening processes (Sachdev & Gupta, 2019). In material science, feature interactions reveal relationships that enable the development of innovative materials with desired properties (Apel et al., 2013; Liu et al., 2023b). In business, analyzing interactions between customer demographics and purchasing behaviors can inform personalized marketing strategies and enhance customer retention (Liu et al., 2023a). In genomics, interactions between genes or "epistasis", is believed to play a major role in driving complex diseases such as asthma, diabetes, multiple sclerosis, and various cardiovascular diseases (Cordell, 2002; Phillips, 2008; Mackay & Moore, 2014; Li et al., 2020; Guindo-Martínez et al., 2021; Zeng et al., 2022; Singhal et al., 2023; Wang et al., 2023). These application areas underscore the need for machine learning tools that can accurately detect and quantify such feature interactions, providing interpretable and actionable insights for potentially high-dimensional data.

In addition to detecting feature interactions, it is crucial to quantify the uncertainty associated with these interactions to ensure they are not simply noise. Uncertainty quantification has emerged as a focus in machine

learning, addressing the need for more reliable and interpretable models (Lei et al., 2018; Tibshirani et al., 2019; Barber et al., 2021; Romano et al., 2019; Kim et al., 2020; Chernozhukov et al., 2021). While much of this work has traditionally centered on quantifying uncertainty in predictions (Lei et al., 2018; Barber et al., 2021), there has been a growing interest in extending these techniques to feature importance (Rinaldo et al., 2019; Zhang & Janson, 2020; Zhang et al., 2022; Dai et al., 2022; Williamson et al., 2023). However, uncertainty quantification for feature interactions remains largely unexplored.

Motivated by these challenges, we introduce the interaction Leave-One-Covariate-Out (iLOCO) metric and inference framework, which addresses key limitations of existing methods. Our framework provides a model-agnostic, assumption-light, statistically and computationally efficient solution for quantifying feature interactions and their uncertainties across diverse applications.

## 1.1 Related Works

While feature importance has been extensively studied (Samek et al., 2021; Altmann et al., 2010; Lipovetsky & Conklin, 2001; Murdoch et al., 2019), work on feature interactions remains relatively limited. Shapley-based approaches, such as FaithSHAP, extend the game-theoretic framework to quantify pairwise interactions by decomposing predictions into main and interaction effects (Tsai et al., 2023; Sundararajan et al., 2020; Rabitti & Borgonovo, 2019). The H-statistic, based on partial dependence, measures the proportion of output variance attributable to interactions (Friedman & Popescu, 2008). Although both of these methods are model-agnostic methods and theoretically grounded, their computational cost increases exponentially with the number of features and observations, limiting their scalability. Recent work has also begun exploring interactions in large language models (LLMs), using attribution techniques to understand how combinations of input tokens influence prediction (Kang et al., 2025). Model-specific alternatives have been developed for certain model classes, such as Iterative Forests for random forests (Basu et al., 2018; Behr et al., 2022), Integrated Hessians for neural networks (Janizek et al., 2021), and linear regression-based methods (Li et al., 2022; Cordell, 2009; Purcell et al., 2007). Several local interaction detection/attribution methods have also been proposed (Tsang et al., 2020; 2018), focusing on explanations for individual instances instead of global interpretation.

Furthermore, there has been very limited literature on the uncertainty quantification for feature interactions, despite the growing body of research on model-agnostic feature importance inference (Lei et al., 2018; Zhang et al., 2022; Williamson et al., 2023; Williamson & Feng, 2020; Abdar et al., 2021; Grigoryan & Collins, 2023; Shah & Peters, 2020; Watson & Wright, 2021). For example, bootstrap confidence intervals have been discussed and implemented for Accumulated Local Effects (ALE) (Apley & Zhu, 2016; Okoli, 2023), which provide curve- or surface-level summaries of main effects and low-order interactions for a fixed trained model. Such procedures are valuable for visualization, but they target ALE effect curves and surfaces rather than predictive-risk-based global interaction measures designed for interaction detection. In addition, existing bootstrap intervals for ALE are mainly used as practical uncertainty summaries, rather than a general inference framework with formal validity guarantees. For instance, since ALE is estimated on a finite grid, often with data-dependent bins, classical bootstrap theory may apply to the discretized estimator with fixed bins but would not by itself imply valid inference for the original integral-form ALE functional. For Shapley-value-based interactions, existing variance estimates often quantify Monte Carlo approximation error from sampling coalitions in the combinatorial Shapley formula (Fumagalli et al., 2023), which is distinct from finite-sample statistical uncertainty.

These limitations motivate the development of scalable metrics that can be applied across a wide range of models, associated with rigorous statistical inference procedures. Our goal is to develop confidence intervals for our feature interaction metric that reflect its statistical significance and uncertainty. Our method builds upon the Leave-One-Covariate-Out (LOCO) metric and inference framework originally proposed by (Lei et al., 2018) and later studied by (Rinaldo et al., 2019; Gan et al., 2022). Since computational cost is a major challenge for interaction methods, we improve efficiency by adopting a fast inference strategy based on minipatch (Gan et al., 2022; Yao et al., 2021; Toghani & Allen, 2021) ensembles, which simultaneously subsample both observations and features.

## 1.2 Contributions

Our work focuses on two main contributions. First, we introduce iLOCO, a model-agnostic method for quantifying feature interactions that can be applied to any predictive model without relying on assumptions about the underlying data distribution. Moreover, we introduce an efficient way of estimating this metric via minipatch ensembles. Second, we develop rigorous assumption-light inference procedures for interaction effects, enabling asymptotically valid confidence intervals for interactions detected without making strong distributional assumptions. Collectively, our contributions offer notable advancements in quantifying feature interaction importance and even higher-order interactions in interpretable machine learning.

# 2 iLOCO

## 2.1 Review: LOCO Metric

To begin, let us first review the widely used LOCO metric that is used for quantifying individual feature importance in predictive models (Lei et al., 2018). For a feature $j$, LOCO measures its importance by evaluating the change in prediction error when the feature is excluded. Formally, given a prediction function $f : \mathcal{X} \to \mathbb{R}$ and an error measure $\mathrm{Err}()$, the LOCO importance of feature $j$ is:

$$\Delta_j(\mathbf{X}, \mathbf{Y}) = \mathbb{E}\left[\mathrm{Err}(Y, f^{-j}(X^{-j}; \mathbf{X}^{-j}, \mathbf{Y})) - \mathrm{Err}(Y, f(X; \mathbf{X}, \mathbf{Y})) \,\middle|\, \mathbf{X}, \mathbf{Y}\right], \tag{1}$$

where $f^{-j}$ is the prediction function trained using data $(\boldsymbol{X}, \boldsymbol{Y})$ without feature $j$. A large positive $\Delta_j$ value suggests the importance of the feature by indicating performance degradation when the feature is excluded. While LOCO is effective at measuring the importance of individual features, it fails to capture feature interactions. To address this challenge, we propose the interaction LOCO (iLOCO) metric that extends LOCO to explicitly account for pairwise feature interactions.

## 2.2 iLOCO Metric

Inspired by LOCO, we seek to define a score that measures the influence of the interaction of two variables $j$ and $k$. Consider the expected difference in error between a further reduced model $f^{-(j,k)}$ and the full model:

$$\Delta_{j,k}(\mathbf{X}, \mathbf{Y}) = \mathbb{E}\left[\mathrm{Err}(Y, f^{-(j,k)}(X^{-(j,k)}; \mathbf{X}^{-(j,k)}, \mathbf{Y})) - \mathrm{Err}(Y, f(X; \mathbf{X}, \mathbf{Y})) \,\middle|\, \mathbf{X}, \mathbf{Y}\right] \tag{2}$$

The reduced model $f^{-(j,k)}$ removes covariates $j$ and $k$, their pairwise interaction, and any group involving $j$ or $k$. The quantity $\Delta_{j,k}$ captures the predictive power contributed by $j, k$ and all interactions that include them. This naturally leads to our iLOCO metric:

**Definition 1.** *For two features $j, k$, the iLOCO metric is defined as:*

$$\mathrm{iLOCO}_{j,k} = \Delta_j + \Delta_k - \Delta_{j,k}. \tag{3}$$

Notice this can also be written as

$$\mathrm{iLOCO}_{j,k} = \mathbb{E}\left[\mathrm{Err}(Y, f^{-j}(X^{-j}; \mathbf{X}^{-j}, \mathbf{Y})) + \mathrm{Err}(Y, f^{-k}(X^{-k}; \mathbf{X}^{-k}, \mathbf{Y}))\right.$$
$$\left. - \mathrm{Err}(Y, f^{-(j,k)}(X^{-(j,k)}; \mathbf{X}^{-(j,k)}, \mathbf{Y})) - \mathrm{Err}(Y, f(X; \mathbf{X}, \mathbf{Y})) \,\middle|\, \mathbf{X}, \mathbf{Y}\right] \tag{4}$$

which highlights that iLOCO compares the total effect of removing features $j$ and $k$ individually versus removing them jointly. The final subtraction of $\mathrm{Err}(Y, f(X; \mathbf{X}, \mathbf{Y}))$ corrects for double-counting the full model's baseline error, which appears in both $\Delta_j$ and $\Delta_k$. Without this correction, any overlapping contribution of $j$ and $k$ would be inflated. A large positive iLOCO value suggests that $j$ and $k$ work together to improve predictive accuracy beyond their individual effects.

To validate and theoretically examine what the iLOCO score captures, we adopt a framework commonly used in the interpretable machine learning literature to examine the decomposition of feature contributions: a functional ANOVA decomposition (Hoeffding, 1948; Stone, 1994; Hooker, 2004; 2007).

**Assumption 1.** *We assume that the conditional mean function $f^*(X) = \mathbb{E}[Y \mid X]$ admits a functional ANOVA decomposition of the form*

$$f^*(X) = g_0 + \sum_{j=1}^{M} g_j(X_j) + \sum_{j<k} g_{j,k}(X_j, X_k) + \sum_{j<k<l} g_{j,k,l}(X_j, X_k, X_l) + \cdots = \sum_{u \subseteq [M]} g_u(X_u),$$

*where $[M]$ is the index set of the feature space $\mathcal{X}$, and each function component above satisfies $\mathbb{E}[g_u(X_u)] = 0$ and $\mathbb{E}[g_u(X_u)g_v(X_v)] = 0$ whenever $u \neq v$.*

Note that much of the work on functional ANOVA assumes orthogonality of the functions and Assumption 1 can be viewed as a probabilistic extension of such conditions Hooker (2007). We can show that Assumption 1 is always satisfied when features are independent; more discussion and justification for Assumption 1 is included in the supplemental material. For illustrative purposes, suppose we replace the fitted models in the iLOCO metric by their population counterparts. The resulting theoretical version of the iLOCO score is defined as $\text{iLOCO}^*_{j,k} = \Delta^*_j + \Delta^*_k - \Delta^*_{j,k}$, where each term quantifies the expected increase in prediction error when specific subsets of features are removed from the population model. For example, $\Delta^*_j = \mathbb{E}[\text{Err}(Y, f^{*-j}(X^{-j}))] - \mathbb{E}[\text{Err}(Y, f^*(X))]$, where $f^{*-j}$ removes all $g_u$ terms with $j \in u$ in the functional ANOVA decomposition. Analogous definitions hold for $\Delta^*_k$ and $\Delta^*_{j,k}$.

**Proposition 1.** *Suppose Assumption 1 holds. Then:*

(a) ***Regression*** *Suppose that $Y = f^*(X) + \epsilon$ with $\epsilon$ being zero-mean random noise of finite second moment, independent from $X$. If $\text{Err}(Y, \hat{Y}) = (Y - \hat{Y})^2$, then $\text{iLOCO}^*_{j,k} = \sum_{u \subseteq [M]:j,k \in u} \mathbb{E}[g_u^2(X_u)]$.*

(b) ***Classification*** *If $Y \sim \text{Bernoulli}(f^*(X))$ and $\text{Err}(Y, \hat{Y}) = |Y - \hat{Y}|$, then $\text{iLOCO}^*_{j,k} = 2\sum_{u \subseteq [M]:j,k \in u} \mathbb{E}[g_u^2(X_u)]$.*

This result provides a clear interpretation for our iLOCO metric, demonstrating that when (i) Assumption 1 holds (implied by independent features) and (ii) the prediction functions are the population models, $\text{iLOCO}_{j,k}$ captures the variance contribution arising specifically from the joint interaction terms of $j$ and $k$, including higher-order interactions. Similar to the LOCO metric, feature correlations and estimation errors of the employed machine learning model for the population model can also lead to deviation of the iLOCO metric from the clean representation above. Interestingly, as will be discussed in Section 2.5, another potential usage of iLOCO might be a promising tool for handling correlated features. Full derivations and proofs of Proposition 3 are provided in the supplemental material.

### 2.3 iLOCO Estimation

The iLOCO metric defined in equation 3 involves fitted prediction functions $f$, $f^{-j}$, $f^{-k}$, $f^{-(j,k)}$ trained from $(\boldsymbol{X}, \boldsymbol{Y})$ and an expectation over test point $(X, Y)$. In practice, we do not have access to the expectation over $(X, Y)$, but we can estimate it using new data samples. To this end, we introduce two estimation strategies, iLOCO via data-splitting and iLOCO via minipatch ensembles; the latter is specifically tailored to address the major computational burden that arises from fitting models leaving out all possible interactions.

**iLOCO via Data Splitting**
The data-splitting approach estimates iLOCO by partitioning the dataset $(\mathbf{X}, \mathbf{Y})$ into a training set $D_1 = (\mathbf{X}^{(1)}, \mathbf{Y}^{(1)})$ and a test set $D_2 = (\mathbf{X}^{(2)}, \mathbf{Y}^{(2)})$. We train the full model $\hat{f}$ along with the models excluding $j$, $k$, and $j, k$ on training dataset $D_1$. The error functions for the full model and the corresponding feature-excluded error functions are evaluated on the test set $D_2$. This approach effectively computes iLOCO, but it comes with certain challenges. Since it involves training multiple models and only utilizes a random subset of the dataset for each, there may be concerns about computational efficiency and the stability of the resulting metric.

**iLOCO via Minipatches**
To address the computational and statistical limitations of the data splitting approach above, we introduce iLOCO-MP (minipatches), an efficient estimation method that leverages ensembles of subsampled observations

and features. Let $N$ denote the total number of observations and $M$ the number of features in the dataset. This method repeatedly constructs minipatches by randomly sampling $n$ observations and $m$ features from the full dataset (Yao & Allen, 2020). For each minipatch, a model is trained on the reduced dataset, and we can evaluate predictions on left-out observations. Specifically, for features $j$ and $k$, the leave-one-(observation)-out and leave-two-covariates-out prediction is defined as $\hat{f}_{-i}^{-(j,k)}(X_i) = \frac{1}{\sum_{b=1}^{B} \mathbb{I}(i \notin I_b)\mathbb{I}(j,k \notin F_b)} \sum_{b=1}^{B} \mathbb{I}(i \notin I_b)\mathbb{I}(j,k \notin F_b)\hat{f}_b(X_i)$, where $I_b \subset [N]$ is the set of observations selected for the $b$-th minipatch, $F_b \subset [M]$ is the set of features selected, $\hat{f}_b$ is the model trained on the minipatch, and $\mathbb{I}(\cdot)$ is an indicator function that checks whether an index is excluded. This formulation ensures that the predictions exclude both the observation $i$ and the features $j$ and $k$. For each sample $i$, we can compute the other leave-$i$-out predictions accordingly and compare their prediction errors; averaging the prediction error differences over all samples leads to an estimate for iLOCO. We note that after fitting minipatch ensembles, computing the iLOCO metric requires no further model fitting and is nearly free computationally. Furthermore, avoiding data-splitting ensures a stable approximation of the iLOCO metric. The full iLOCO-MP algorithm is given in the supplement.

### Distinction between the iLOCO Metrics Estimated Above

One clarification we would like to make is that the iLOCO metric is defined with respect to a particular collection of full and reduced models, e.g., $f$ and $f^{-(j,k)}$. Therefore, when different models are trained, the corresponding iLOCO metric also changes. The data-splitting approach estimates the iLOCO metric associated with models trained by any chosen algorithm on one the training set $D_1$. In contrast, the minipatch approach estimates the iLOCO metric associated with predictors that take the form of minipatch ensembles, built using any base learner and trained using the entire data set. Therefore, iLOCO-Split is conceptually straightforward and can be applied to arbitrary fitted models, whereas iLOCO-MP avoids refitting large models for each feature, uses the full dataset more efficiently, and provides a convenient, nearly cost-free post-training inference procedure for minipatch ensembles.

### 2.4 Extension to Higher-Order Interactions

Detecting higher-order feature interactions is crucial in understanding complex relationships in data, as many real-world phenomena involve intricate dependencies among multiple features that cannot be captured by pairwise interactions alone. To extend the iLOCO metric to account for such higher-order interactions, we generalize its definition to isolate the unique contributions of $S$-way interactions among features:

**Definition 2.** *For an $S$-way interaction, the iLOCO metric is defined as:*

$$\text{iLOCO}_S = \sum_{T \subseteq S, T \neq \emptyset} (-1)^{|T|+1} \Delta_T.$$

Here, we sum over all possible subsets of $S$, and for each subset $T$, $\Delta_T$ denotes the contribution of $T$ to the model error. The term $(-1)^{|T|+1}$ alternates the sign based on the size of $T$, ensuring proper aggregation and cancellation of irrelevant terms that appear multiple times. In the supplemental material, we have a generalized version of Proposition 1 that justifies the S-way interaction iLOCO score; in a nutshell, under Assumption 1, the $\text{iLOCO}_S$ sums over the squared norm of the function terms $g_u$ where $u \supseteq S$. By leveraging this generalized formulation, iLOCO extends its ability to capture and quantify the contributions of higher-order interactions, providing a more complete understanding of complex feature relationships.

In practice, the higher-order iLOCO metric can be estimated using the same approaches described in Section 2.3. However, higher-order iLOCO estimation also poses a non-trivial computational challenge. Under the data-splitting approach, estimating $\Delta_T$ for all nonempty subsets $T \subset S$ requires fitting a total of $2^{|S|} - 1$ models, so the computational cost grows exponentially with the interaction order. For the minipatch approach, fitting each minipatch is simple, and the method scales much better than the data-splitting approach for pairwise interactions. Nevertheless, higher-order interactions remain challenging. Although minipatch ensembles avoid the need to explicitly refit each reduced model $f^{-T}$, the number of minipatches may need to grow exponentially in $|S|$ under uniform sampling to ensure that each subset $T \subset S$ is adequately sampled and that $\Delta_T$ can consequently be estimated with sufficient accuracy. Therefore, pairwise interactions are the primary practical target of the current methods, while low-order extensions, such as third-order interactions,

may be feasible in moderate-scale settings. We expect that combining the minipatch method with a carefully designed adaptive sampling procedure could help address this computational challenge, and we leave this as an important direction for future work.

## 2.5 iLOCO for Correlated, Important Features

Feature correlation is a known, unsolved challenge in interpretable machine learning. The independent feature assumption is often made to establish theoretical guarantees for feature interaction recovery (Behr et al., 2022); even for individual feature importance, many have shown the distortion of the importance metrics under correlation (Verdinelli & Wasserman, 2024);

In particular, the feature correlation issue can be especially pronounced for LOCO feature importance because when correlated features are removed individually, the remaining correlated feature(s) can partially compensate, resulting in a small LOCO metric that can miss important but correlated features. Interestingly, our proposed iLOCO metric may have the potential of addressing this problem. Note that for a strongly correlated and important feature pair $j$ and $k$, $\Delta_j$ and $\Delta_k$ (and LOCO) will both be near zero as they compensate for each other. But, $\Delta_{j,k}$ will have a strong positive effect, and thus our iLOCO metric will be strongly negative for important, correlated feature pairs. Our iLOCO metric may therefore serve dual purposes: positive values indicate an important pairwise interaction whereas negative values may indicate individually important but correlated feature pairs, an observation worth noting as a potential additional usage of the iLOCO metric. This alternative application may also be a promising strategy for addressing the correlated feature challenge in feature importance estimation and inference, and could further extend to higher-order iLOCO metrics to help alleviate the correlation challenge in feature interaction detection. We leave a more thorough investigation of these ideas for future work.

# 3 Statistical Inference for iLOCO with Assumption-light Validity

Beyond just detecting interactions via our iLOCO metric, it is critical to quantify the uncertainty in these estimates to understand if they are statistically significant. To address this, we develop confidence intervals for both our iLOCO-Split and iLOCO-MP estimators that are asymptotically valid without making strong assumptions on the data generating distribution or the employed machine learning model.

## 3.1 iLOCO Inference via Data Splitting

As previously outlined, we can estimate the iLOCO metric via data splitting, where the training set $D_1$ is used for training predictive models $\hat{f}$, $\hat{f}^{-j}$, $\hat{f}^{-k}$, $\hat{f}^{-(j,k)}$, while the test set $D_2$ with $N^{\mathcal{D}_{\text{test}}}$ samples is used for evaluating the error functions for each trained model. Each test sample gives an unbiased estimate of the target iLOCO metric, $\text{iLOCO}_{j,k}^{\text{split}} = \Delta_j^{\text{split}} + \Delta_k^{\text{split}} - \Delta_{j,k}^{\text{split}}$, where $\Delta_{j,k}^{\text{split}} = \mathbb{E}[\text{Err}(Y, \hat{f}^{-(j,k)}(X)) - \text{Err}(Y, \hat{f}(X))|\boldsymbol{X}^{(1)}, \boldsymbol{Y}^{(1)}]$, and $\Delta_j^{\text{split}}$, $\Delta_k^{\text{split}}$ are defined similarly. Here, the expectation is taken over the unseen test data, conditioning on $D_1$, which is a slightly different inference target than defined in equation 3 which conditions on the model trained on all available data. To perform statistical inference for $\text{iLOCO}_{j,k}^{\text{split}}$, we follow the approach of Lei et al. (2018), collecting its estimates on all test samples $\{\widehat{\text{iLOCO}}_{j,k}(X_i^{(2)}, Y_i^{(2)})\}_{i=1}^{N^{\mathcal{D}_{\text{test}}}}$ and constructing a confidence interval $\hat{\mathbb{C}}_{j,k}^{\text{split}}$ based on a normal approximation. The detailed inference algorithm for iLOCO-Split is included in the appendix. To assure valid asymptotic coverage of $\hat{\mathbb{C}}_{j,k}^{\text{split}}$ for inference target $\text{iLOCO}_{j,k}^{\text{split}}$, we need the following assumption:

**Assumption 2.** *Assume a bounded standthird moment:* $\mathbb{E}(\widehat{\text{iLOCO}}_{j,k}(X_i^{(2)}, Y_i^{(2)}) - \text{iLOCO}_{j,k}^{\text{split}})^3/\sigma_{j,k}^3 \leq C$, *where* $\sigma_{j,k}^2 = \text{Var}(\widehat{\text{iLOCO}}_{j,k}(X_i^{(2)}, Y_i^{(2)})|\boldsymbol{X}^{(1)}, \boldsymbol{Y}^{(1)})$.

**Theorem 1** (Coverage of iLOCO-Split). *Suppose Assumption 2 holds and* $\{(X_i, Y_i)\}_{i=1}^N$ *are i.i.d., then we have* $\lim_{N^{\mathcal{D}_{\text{test}}} \to \infty} \mathbb{P}(\text{iLOCO}_{j,k}^{\text{split}} \in \mathbb{C}_{j,k}^{\text{split}}) = 1 - \alpha$.

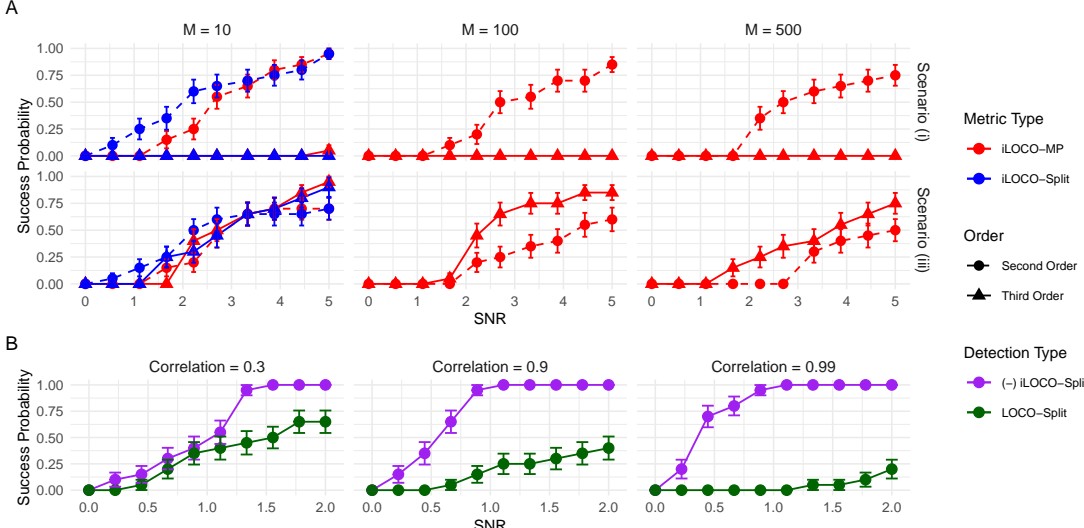

Figure 1: **Validation of iLOCO Metric.** Part A shows the success probability of identifying an interaction pair in nonlinear classification scenarios (i) and (iii) using an MLP classifier. Part B presents the success probability of detecting an important, correlated feature pair across varying correlation strengths.

### 3.2 iLOCO Inference via Minipatches

Recall that after fitting minipatch ensembles, estimating the iLOCO metric is computationally free, as estimates aggregate over left out observations and/or features. Such advantages also carry over to statistical inference. In particular, we aim to perform inference for the target iLOCO score defined in equation 3 with predictive models (e.g., $f$, $f^{-j}$) all being minipatch ensembles. This inference target is denoted by $\text{iLOCO}_{j,k}^{\text{MP}}$, and it is conditioned on all data instead of a random data split, $D_1$ as with iLOCO-Split. Then, for each sample $i$, we can compute the leave-one-observation-out predictors $\hat{f}_{-i}$, $\hat{f}_{-i}^{-j}$, $\hat{f}_{-i}^{-k}$, $\hat{f}_{-i}^{-(j,k)}$ simply by aggregating appropriate minipatch predictors, and evaluating these predictors on sample $i$ to obtain $\widehat{\text{iLOCO}}_{j,k}(X_i, Y_i)$. For statistical inference, we collect $\{\widehat{\text{iLOCO}}_{j,k}(X, Y)\}_{i=1}^N$ and construct a confidence interval $\mathbb{C}_{j,k}^{\text{MP}}$ based on a normal approximation. The detailed inference procedure is given in the appendix.

Despite the fact that $\widehat{\text{iLOCO}}_{j,k}(X_i, Y_i)$ has a complex dependency structure since all the data is essentially used for both fitting and inference, we show asymptotically valid coverage of $\mathbb{C}_{j,k}^{\text{MP}}$ under some mild assumptions. First, let $h_{j,k}(X, Y)$ be the interaction importance score of feature pair $(j, k)$ evaluated at sample $(X, Y)$, with expectation taken over the training data $(\boldsymbol{X}, \boldsymbol{Y})$ that gives rise to the predictive models; let $(\sigma_{j,k}^{\text{MP}})^2 = \text{Var}(h_{j,k}(X_i, Y_i))$.

**Assumption 3.** *Assume a bounded third moment: $\mathbb{E}[h_{j,k}(X, Y) - \mathbb{E}h_{j,k}(X, Y)]^3 / (\sigma_{j,k}^{\text{MP}})^3 \leq C$.*

**Assumption 4.** *The error function is Lipschitz continuous w.r.t. the prediction: for any $Y \in \mathbb{R}$ and any predictions $\hat{Y}_1, \hat{Y}_2 \in \mathbb{R}^d$, $|\text{Err}(Y, \hat{Y}_1) - \text{Err}(Y, \hat{Y}_2)| \leq L\|\hat{Y}_1 - \hat{Y}_2\|_2$.*

Common error functions like mean absolute error trivially satisfy this assumption.

**Assumption 5.** *The prediction difference between the predictors trained on different minipatches are bounded by $D$ at any input value $X$.*

**Assumption 6.** *The minipatch sizes $(m, n)$ satisfy $\frac{m}{M}$, $\frac{n}{N} \leq \gamma$ for some constant $0 < \gamma < 1$, and $n = o(\frac{\sigma_{j,k}^{\text{MP}}}{LD}\sqrt{N})$.*

**Assumption 7.** *The number of minipatches satisfies $B \gg (\frac{D^2 L^2 N}{(\sigma_{j,k}^{\text{MP}})^2} + 1)\log N$.*

These are mild assumptions on the minipatch size and number. Further, requiring the predictions from any pair of minipatches must simply be bounded, is a much weaker condition than the stability conditions typical

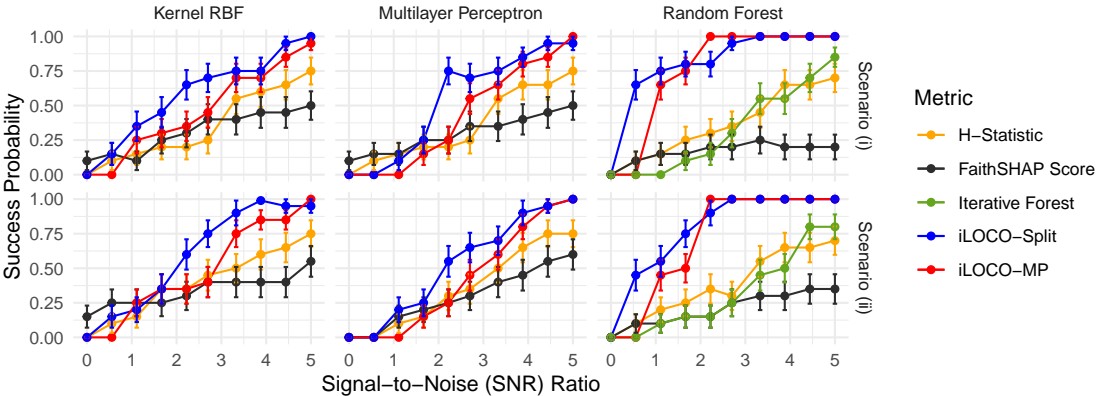

Figure 2: **Comparative Evaluations.** Success probability of detecting feature pair $(1, 2)$ across SNR levels for KRBF, RF, and MLP classifiers on nonlinear classification simulations (i) and (ii).

in the leave-one-out distribution-free inference literature (Barber et al., 2019). Instead, the mild boundedness assumption helps us establish the stability for the whole ensemble and consequently the validity of iLOCO inference.

**Theorem 2** (Coverage of iLOCO-MP). *Suppose Assumptions 3-7 hold and $\{(X_i, Y_i)\}_{i=1}^N$ i.i.d., then we have $\lim_{N\to\infty} \mathbb{P}(\text{iLOCO}_{j,k}^{\text{MP}} \in \mathbb{C}_{j,k}^{\text{MP}}) = 1 - \alpha$.*

The detailed theorems, assumptions, and proofs can be found in the appendix. Note that the proof follows closely from that of (Gan et al., 2022), with the addition of a third moment condition to imply uniform integrability. Overall, our work has provided the first model-agnostic inference framework for LOCO-type feature interactions that is asymptotically valid. Notably, we do not make any strong assumptions on the data-generating model or the implemented base machine learning model, other than some mild conditions such as boundedness (Assumption 5). Further, note that while iLOCO inference via minipatches requires utilizing minipatch ensemble predictors, it gains in both statistical and computational efficiency as it does not require data splitting and conducting inference conditional on only a random portion of the data.

## 4 Empirical Studies

### 4.1 Simulation Setup and Results

We design simulation studies to validate the proposed iLOCO metric and its inference procedure. We organize our empirical studies around three goals: (i) validating that the iLOCO metric correctly recovers true feature interactions across a range of signal strengths, dimensionalities, and model classes; (ii) comparing iLOCO against existing interaction detection methods; and (iii) confirming that the confidence intervals produced by iLOCO-Split and iLOCO-MP achieve valid empirical coverage.

Data are generated as $X_i \overset{i.i.d}{\sim} N(0, \mathbf{I})$ with $M = 10$ features and $N = 500$ samples in the base case. We consider three scenarios spanning classification/regression and linear/nonlinear settings: (i) $f(\mathbf{X}) = snr \cdot (X_1 X_2) + \mathbf{X}\boldsymbol{\beta}$; (ii) $f(\mathbf{X}) = snr \cdot (X_1 X_2) + X_2 X_3 + X_3 X_4 + X_4 X_5 + \mathbf{X}\boldsymbol{\beta}$; and (iii) $f(\mathbf{X}) = snr \cdot (X_1 X_2 X_3) + \mathbf{X}\boldsymbol{\beta}$. Here, $\beta_j \sim N(2, 0.5)$ for $j = 1, \dots, 5$ and $\beta_j = 0$ for $j = 6, \dots, 10$. Since we focus on detecting the $(1, 2)$ interaction, the scalar $snr$ controls its signal strength. For nonlinear variants, we apply a tanh transformation to each interaction term. In regression, $Y = f(\mathbf{X}) + \epsilon$ with $\epsilon \sim N(0, 1)$; in classification, $Y \sim \text{Bern}(\sigma(f(\mathbf{X})))$, where $\sigma$ is the sigmoid function. We set the number of minipatches for iLOCO-MP to $B = 10,000$ with minipatch sizes $m = 20\%M$, $n = 20\%N$; however, for the simulations in Figure 1A, we increase the sample size to $N = 10,000$, vary the feature dimensionality with $M \in \{10, 100, 500\}$, and use $B = 200,000$ to stabilize inference. For iLOCO-Split, we split the data 50/50 between training and calibration. Throughout our experiments, except for the validation of coverage, error bars represent 95% confidence intervals centered at the mean iLOCO estimate with width determined by the estimated standard deviation of the per-sample scores. The coverage curves in Figure 3 report empirical coverage computed by repeating this estimation and

interval-construction procedure across 50 independent replicates.. We compare against baselines including the H-Statistic (Friedman & Popescu, 2008), FaithSHAP (Tsai et al., 2023), and Iterative Forests (Basu et al., 2018) and utilize the corresponding available code as written. Performance is evaluated via success probability, the proportion of times the $(1, 2)$ pair receives the highest interaction score, allowing consistent comparisons across methods and relative calibration of feature importance. Results are shown for three model classes: kernel support vector machines with radial basis function kernels (KRBF), multilayer perceptrons (MLP), and random forests (RF). A fixed set of hyperparameters for each model was chosen via cross-validation. The MLP model uses ReLU activiation with a single hidden layer in all experiments except Figure 1A, where a three-hidden-layer architecture is used.

We first validate that iLOCO reliably recovers true feature interactions across varying signal strengths and feature dimensionalities. Figure 1 evaluates its ability to recover true feature interactions under varying signal-to-noise ratios (SNRs) and feature dimensionalities for an MLP classifier. Part (A) reports the success probability of identifying the true interacting pair as SNR increases . Rows correspond to interaction structures: scenario (i) with a pairwise interaction and scenario (iii) with a tertiary interaction. Columns reflect increasing feature dimensionality. The results show that second-order iLOCO reliably identifies the true pairwise interaction in scenario (i). It can also identify the tertiary interaction in scenario (iii), but with diminished performance compared to the third-order iLOCO which is specifically designed to detect the tertiary interaction. These results align with our theory, thus validating our metric.

Part (B) assesses detection under feature correlation. Success for iLOCO is defined as correctly identifying the correlated and important pair (1,2) as the pair with the most negative value, while success for LOCO is defined as correctly identifying features 1 and 2 as the top two features As we expect, the LOCO metric breaks down with high correlation, but our negative iLOCO nicely detects important but correlated features, thus potentially solving an open and challenging problem in feature importance. Additional results for a RF classifier can be found in the supplement.

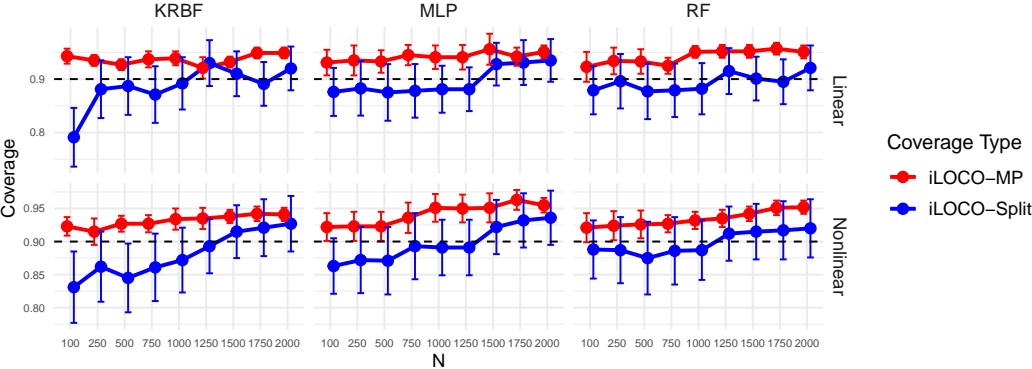

Figure 3: **Theory Validation.** Coverage of 90% confidence intervals for a null feature pair in synthetic regression simulation (i) using KRBF, MLP, and RF as the base estimators.

We then evaluate iLOCO's performance relative to existing feature interaction detection methods across multiple model classes and simulation scenarios.. Figure 2 compares the success probability of detecting the true interacting feature pair (1,2) across increasing SNR levels for various interaction detection methods applied to nonlinear classification scenarios. Results are shown for three model classes, KRBF, MLP, RF, across scenarios (i) (top) and (ii) (bottom). In both scenarios, the pair (1,2) carries signal, so success probability is expected to rise with increasing SNR. Across all model classes and scenarios, iLOCO-Split (blue) and iLOCO-MP (red) consistently outperform baseline methods, especially at higher SNRs. Notably, FaithSHAP (black) exhibits inflated success probabilities at low SNR, suggesting poor calibration and potential spurious detection of interactions. These findings demonstrate the robustness and accuracy of iLOCO across diverse model types and data settings. Additional simulation results for linear and nonlinear, classification and regression, and correlated features with $\mathbf{\Sigma} \neq \mathbf{I}$ are in the supplemental material.

Table 1: Timing results (seconds) for various dataset sizes using Simulation 1 and the KRBF regressor for all methods except Iterative Forest, where RF regressor was used. As $M$ and $N$ grow, computing interaction importance scores using H-Statistic and Faith Shap becomes infeasible. Note that the (p) indicates the code for that method was distributed across multiple processes.

| Method | $N = 250, M = 10$ | $N = 500, M = 20$ | $N = 1000, M = 100$ | N = 10000, M = 500 |
|---|---|---|---|---|
| H-Statistic | 285.4 | 97201.2 | > 6 days | > 6 days |
| Faith-Shap | 70.2 | 72801.3 | Not Supported | Not Supported |
| Iterative Forest | 14.1 | 17.8 | 20.3 | 76.8 |
| iLOCO-Split | 16.8 (p) | 144.7 (p) | 738.3 (p) | 5481.2 (p) |
| iLOCO-MP | 24.3 (p) | 27.2 (p) | 38.7 (p) | 193.5 (p) |

Additionally, we construct an empirical study to demonstrate that the interaction feature importance confidence intervals generated by iLOCO-MP and iLOCO-Split have valid coverage for the inference target. We compute coverage by evaluating the respective inference targets of iLOCO-Split and iLOCO-MP via Monte Carlo estimates over 10,000 new test points, conditioning on the training set for iLOCO-Split and on the full dataset for iLOCO-MP. For iLOCO-MP, we use $B = 10,000$ random minipatches. We evaluate the coverage of the confidence intervals constructed from 50 replicates. Figure 3 shows coverage rates for iLOCO-Split and iLOCO-MP of 90% confidence intervals for a null feature pair in regression simulation (i) using various base estimators. iLOCO-MP exhibits slight over-coverage whereas iLOCO-Split has valid coverage for sufficiently large sample size, as expected with the reduced sample size due to data-splitting and asymptotic coverage results. We include additional studies where SNR $= 2$ in the supplemental material. These results validate our statistical inference theory.

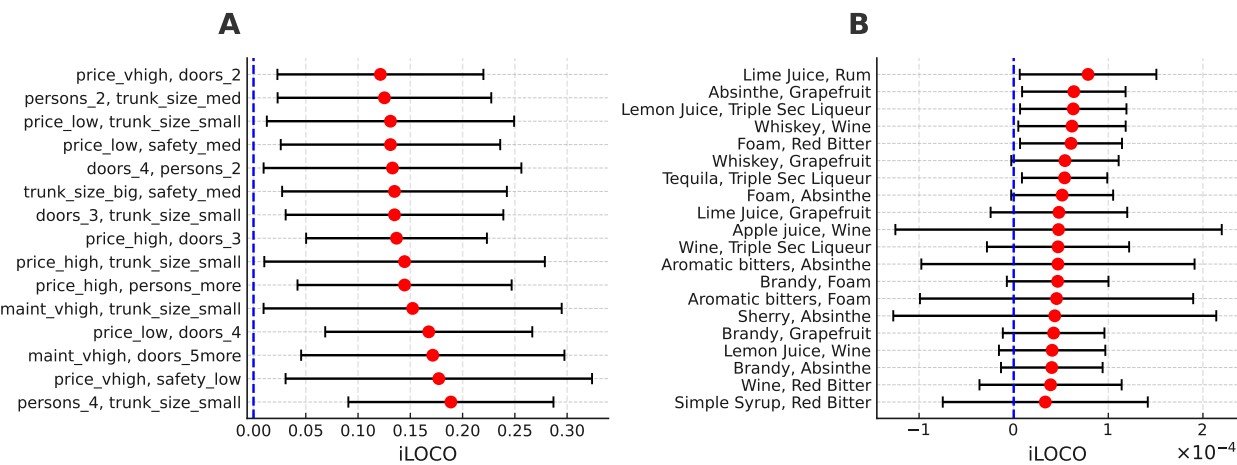

Figure 4: iLOCO marginal confidence intervals ($\alpha = 0.1$; adjusted for multiplicity via Bonferroni) on the Car Evaluation data via iLOCO-Split (A) and Cocktail Recipe data via iLOCO-MP (B). Interactions with confidence intervals that do not contain zero (blue dashed line) are statistically significant.

A key advantage of our iLOCO approach, particularly iLOCO-MP, is its exceptional computational efficiency. Table 1 compares the computational time required to calculate feature interaction metrics across all features in the dataset. Results are recorded on an Apple M2 Ultra 24-Core CPU at 3.24 GHz, 76-Core GPU, and 128GB of Unified Memory and across 11 processes when denoted using (p). As the dataset size ($N$) and the number of features ($M$) increase, methods like H-Statistic and Faith-Shap quickly become infeasible. In contrast, iLOCO-MP shows large efficiency gains, making it scalable to larger datasets without sacrificing performance. While Iterative Forest achieves reasonable computational times, it is restricted to random forest models. The efficiency and model-agnostic nature of iLOCO-MP highlight its practicality for real-world use cases involving complex datasets.

### 4.2 Case Studies

We validate iLOCO by computing feature importance scores and intervals for two datasets. The Car Evaluation dataset with $N = 1728$ and $M = 15$ one-hot-encoded features seeks to predict car acceptability (classification task) Bohanec (1988). Due to the small size of the Car dataset, we compute the scores via iLOCO-Split with a Random Forest as the base model. Second, we gathered a new Cocktail Recipe dataset by web scraping the Difford's Guide website (Diffords, 2025). For our analysis, we consider the top $M = 100$ most frequent one-hot-encoded ingredients for $N = 5,934$ cocktails and the task is to predict the official Difford's Guide ratings (regression task). We fit a 3-hidden-layer MLP regressor with 20,000 minipatches, setting $m = 50\%M$ and $n = 50\%N$. In both cases, we set the error rate $\alpha = 0.1$ and adjust for multiplicity via Bonferroni. In Figure 4, feature interactions with confidence intervals that do not contain zero (blue line) are deemed significant.

In Figure 4A, we present the iLOCO-Split scores with Bonferroni adjusted confidence intervals for all feature pairs in the Car Evaluation dataset. Top-ranked interactions include "high maintenance & 4 persons", "very high maintenance & 2 doors," and "low price & large trunk size." These pairings highlight how the perceived burden of upkeep interacts with seating and storage constraints, as well as the appeal of affordability when combined with ample capacity. For example, a low purchase price paired with a large trunk reflects a desirable balance between cost and practicality, while maintenance demands coupled with limited passenger or trunk space underscore trade-offs that reduce overall acceptability.

Figure 4B shows the top 15 ingredient interactions identified by the iLOCO-MP metric, with confidence intervals indicating estimation uncertainty. Several pairs, such as "Lime Juice & Rum" and "Tequila & Cointreau" have positive scores with intervals excluding zero, reflecting significant interactions aligned with classic cocktails like the Daiquiri and Margarita. Others, including "Foam Agent & Absinthe" and "Brandy & Grapefruit Juice," have intervals that include zero, suggesting weaker or inconsistent effects despite high iLOCO values. These results demonstrate iLOCO 's ability to recover meaningful ingredient pairings while providing calibrated uncertainty estimates.

## 5 Discussion

In this work, we propose iLOCO, a model-agnostic metric to quantify feature interactions. We also develop, to the best of our knowledge, the first inference framework for constructing confidence intervals for global feature interaction measures with formal validity guarantees. Additionally, we propose a computationally fast way to estimate iLOCO and conduct inference using minipatch ensembles, allowing our approach to scale both computationally and statistically to large data sets. Our empirical studies demonstrate the superior ability of iLOCO to detect important feature interactions and highlight the importance of uncertainty quantification in this context. We also briefly discussed additional usage of iLOCO for detecting important correlated features as well as iLOCO for higher-order interactions, but future work could consider these important challenges further. Finally, for increasing numbers of features, detecting and testing pairwise and higher-order interactions becomes a major challenge. Future work could consider adaptive learning strategies, perhaps paired with minipatch ensembles (Yao & Allen, 2020), to focus both computational and statistical efforts on only the most important interactions in a large data set.

Beyond methodological contributions, iLOCO offers practical value across the application domains described in the introduction. In drug discovery, statistically significant iLOCO scores can help researchers prioritize candidate compound pairs for experimental validation, focusing costly screening efforts on chemical property interactions most likely to influence efficacy. In genomics, iLOCO confidence intervals provide a principled way to distinguish true epistatic interactions from noise, which is especially important given the high-dimensional nature of genome-wide association studies. In business applications, practitioners can use significant iLOCO interactions to inform personalized marketing strategies, for example by identifying which combinations of customer demographics and purchasing behaviors jointly drive outcomes of interest. More broadly, the confidence intervals produced by iLOCO allow practitioners to make downstream decisions with a quantified sense of uncertainty, distinguishing interactions that are statistically reliable from those that may reflect sampling variability.

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

# A Appendix

# B Inference Algorithms

The iLOCO estimation and inference algorithms are summarized in Algorithms 1 and 2.

---

**Algorithm 1** iLOCO-Split Estimation and Inference

---

**Input:** Training data $(\mathbf{X}^{(1)}, \mathbf{Y}^{(1)})$, test data $(\mathbf{X}^{(2)}, \mathbf{Y}^{(2)})$, features $j, k$, base learners $H$.

1. Split the data into disjoint training and test sets: $(\mathbf{X}^{(1)}, \mathbf{Y}^{(1)})$, $(\mathbf{X}^{(2)}, \mathbf{Y}^{(2)})$.

2. Train prediction models $\hat{f}$, $\hat{f}^{-j}$, $\hat{f}^{-k}$, $\hat{f}^{-(j,k)}$ on $(\mathbf{X}^{(1)}, \mathbf{Y}^{(1)})$:

$$\hat{f}(X) = H(\mathbf{X}^{(1)}, \mathbf{Y}^{(1)})(X), \quad \hat{f}^{-j}(X) = H(\mathbf{X}^{(1),-j}, \mathbf{Y}^{(1)})(X^{-j}),$$
$$\hat{f}^{-k}(X) = H(\mathbf{X}^{(1),-k}, \mathbf{Y}^{(1)})(X^{-k}), \quad \hat{f}^{-(j,k)}(X) = H(\mathbf{X}^{(1),-(j,k)}, \mathbf{Y}^{(1)})(X^{-(j,k)})$$

3. For the $i$th sample in the test set, compute the feature importance and interaction scores:

$$\hat{\Delta}_j(X_i^{(2)}, Y_i^{(2)}) = \text{Error}(Y_i^{(2)}, \hat{f}^{-j}(X_i^{(2),-j})) - \text{Error}(Y_i^{(2)}, \hat{f}(X_i^{(2)})),$$
$$\hat{\Delta}_k(X_i^{(2)}, Y_i^{(2)}) = \text{Error}(Y_i^{(2)}, \hat{f}^{-k}(X_i^{(2),-k})) - \text{Error}(Y_i^{(2)}, \hat{f}(X_i^{(2)})),$$
$$\hat{\Delta}_{j,k}(X_i^{(2)}, Y_i^{(2)}) = \text{Error}(Y_i^{(2)}, \hat{f}^{-(j,k)}(X_i^{(2),-(j,k)})) - \text{Error}(Y_i^{(2)}, \hat{f}(X_i^{(2)})).$$

4. Calculate iLOCO metric for each test sample $i$:

$$\widehat{\text{iLOCO}}_{j,k}(X_i^{(2)}, Y_i^{(2)}) = \hat{\Delta}_j(X_i^{(2)}, Y_i^{(2)}) + \hat{\Delta}_k(X_i^{(2)}, Y_i^{(2)}) - \hat{\Delta}_{j,k}(X_i^{(2)}, Y_i^{(2)}).$$

5. Let $N^{\mathcal{D}_{\text{test}}}$ be the sample size of the test data. Obtain a $1 - \alpha$ confidence interval for $\text{iLOCO}_{j,k}^{\text{split}}$:

$$\mathbb{C}_{j,k}^{\text{split}} = \left[ \overline{\text{iLOCO}}_{j,k} - \frac{z_{\alpha/2}\hat{\sigma}_{j,k}}{\sqrt{N^{\mathcal{D}_{\text{test}}}}}, \quad \overline{\text{iLOCO}}_{j,k} + \frac{z_{\alpha/2}\hat{\sigma}_{j,k}}{\sqrt{N^{\mathcal{D}_{\text{test}}}}} \right],$$

where

$$\overline{\text{iLOCO}}_{j,k} = \frac{1}{N^{\mathcal{D}_{\text{test}}}} \sum_{i=1}^{N^{\mathcal{D}_{\text{test}}}} \widehat{\text{iLOCO}}_{j,k}(X_i^{(2)}, Y_i^{(2)})$$

$$\hat{\sigma}_{j,k} = \sqrt{\frac{1}{N^{\mathcal{D}_{\text{test}}} - 1} \sum_{i=1}^{N^{\mathcal{D}_{\text{test}}}} \left( \widehat{\text{iLOCO}}_{j,k}(X_i^{(2)}, Y_i^{(2)}) - \overline{\text{iLOCO}}_{j,k} \right)^2}$$

**Output:** $\mathbb{C}_{j,k}^{\text{split}}$

---

---

**Algorithm 2** iLOCO-MP Estimation and Inference

---

**Input:** Training pairs $(\mathbf{X}, \mathbf{Y})$, features $j, k$, minipatch sizes $n, m$, number of minipatches $B$, base learners $H$.

1. Perform Minipatch Learning: for $b = 1, \ldots, B$

   - Randomly subsample $n$ observations $I_b \subset [N]$ and $m$ features $F_b \subset [M]$.
   - Train prediction model $\hat{f}_b$ on $(\mathbf{X}_{I_b, F_b}, \mathbf{Y}_{I_b})$:

   $$\hat{f}_b(X) = H(\mathbf{X}_{I_b, F_b}, \mathbf{Y}_{I_b})(X^{F_b}).$$

2. Obtain predictions:

   **LOO prediction:**

   $$\hat{f}_{-i}(X_i) = \frac{1}{\sum_{b=1}^{B} \mathbb{I}(i \notin I_b)} \sum_{b=1}^{B} \mathbb{I}(i \notin I_b) \hat{f}_b(X_i)$$

   **LOO + LOCO (feature $j$):**

   $$\hat{f}_{-i}^{-j}(X_i^{-j}) = \frac{1}{\sum_{b=1}^{B} \mathbb{I}(i \notin I_b) \mathbb{I}(j \notin F_b)} \sum_{b=1}^{B} \mathbb{I}(i \notin I_b) \mathbb{I}(j \notin F_b) \hat{f}_b(X_i^{-j})$$

   **LOO + LOCO (features $j, k$):**

   $$\hat{f}_{-i}^{-(j,k)}(X_i^{-(j,k)}) = \frac{1}{\sum_{b=1}^{B} \mathbb{I}(i \notin I_b) \mathbb{I}(j, k \notin F_b)} \sum_{b=1}^{B} \mathbb{I}(i \notin I_b) \mathbb{I}(j, k \notin F_b) \hat{f}_b(X_i^{-(j,k)})$$

3. Calculate LOO Feature Occlusion:

   $$\hat{\Delta}_j(X_i, Y_i) = \text{Error}(Y_i, \hat{f}_{-i}^{-j}(X_i^{-j})) - \text{Error}(Y_i, \hat{f}_{-i}(X_i))$$
   $$\hat{\Delta}_k(X_i, Y_i) = \text{Error}(Y_i, \hat{f}_{-i}^{-k}(X_i^{-k})) - \text{Error}(Y_i, \hat{f}_{-i}(X_i))$$
   $$\hat{\Delta}_{j,k}(X_i, Y_i) = \text{Error}(Y_i, \hat{f}_{-i}^{-(j,k)}(X_i^{-(j,k)})) - \text{Error}(Y_i, \hat{f}_{-i}(X_i))$$

4. Calculate iLOCO Metric for each sample $i$:

   $$\widehat{\text{iLOCO}}_{j,k}(X_i, Y_i) = \hat{\Delta}_j(X_i, Y_i) + \hat{\Delta}_k(X_i, Y_i) - \hat{\Delta}_{j,k}(X_i, Y_i)$$

5. Obtain a $1 - \alpha$ confidence interval for $\text{iLOCO}_{j,k}^{\text{MP}}$:

   $$\mathbb{C}_{j,k}^{\text{MP}} = \left[ \overline{\widehat{\text{iLOCO}}}_{j,k} - \frac{z_{\alpha/2} \hat{\sigma}_{j,k}}{\sqrt{N}}, \quad \overline{\widehat{\text{iLOCO}}}_{j,k} + \frac{z_{\alpha/2} \hat{\sigma}_{j,k}}{\sqrt{N}} \right]$$

   where

   $$\overline{\widehat{\text{iLOCO}}}_{j,k} = \frac{1}{N} \sum_{i=1}^{N} \widehat{\text{iLOCO}}_{j,k}(X_i, Y_i)$$

   $$\hat{\sigma}_{j,k} = \sqrt{\frac{1}{N-1} \sum_{i=1}^{N} \left( \widehat{\text{iLOCO}}_{j,k}(X_i, Y_i) - \overline{\widehat{\text{iLOCO}}}_{j,k} \right)^2}$$

**Output:** $\mathbb{C}_{j,k}^{\text{MP}}$

---

## C   Inference Theory for iLOCO-Split

We restate our assumptions and theory in Section 3.1 of the main paper with more details.

Recall our inference target: $\text{iLOCO}_{j,k}^{\text{split}} = \Delta_j^{\text{split}} + \Delta_k^{\text{split}} - \Delta_{j,k}^{\text{split}}$, where $\Delta_{j,k}^{\text{split}} = \mathbb{E}[\text{Err}(Y, \hat{f}^{-(j,k)}(X)) - \text{Err}(Y, \hat{f}(X))|\boldsymbol{X}^{(1)}, \boldsymbol{Y}^{(1)}]$, and $\Delta_j^{\text{split}}, \Delta_k^{\text{split}}$ are defined similarly.

**Assumption 8.** *The normalized iLOCO score on a random test sample satisfies the third moment assumption:*
$\mathbb{E}(\widehat{\text{iLOCO}}_{j,k}(X_i^{(2)}, Y_i^{(2)}) - \text{iLOCO}_{j,k}^{\text{split}})^3/(\sigma_{j,k}^{\text{split}})^3 \leq C, \quad \text{where} \quad (\sigma_{j,k}^{\text{split}})^2 = \text{Var}(\widehat{\text{iLOCO}}_{j,k}(X_i^{(2)}, Y_i^{(2)})|\mathbf{X}^{(1)}, \mathbf{Y}^{(1)})$.

We require Assumption 8 to establish the central limit theorem for the iLOCO metrics evaluated on the test data. Prior theory on LOCO-Split (Rinaldo et al., 2019) does not have this assumption, but they focus on a truncated linear predictor and consider injecting noise into the LOCO scores, which can imply a third moment assumption similar to Assumption 8.

**Theorem 3** (Coverage of iLOCO-Split). *Suppose that data $(X_i, Y_i) \overset{i.i.d.}{\sim} \mathcal{P}$. Under Assumption 8, we have $\lim_{N^{\mathcal{D}_{\text{test}}} \to \infty} \mathbb{P}(\text{iLOCO}_{j,k}^{\text{split}} \in \mathbb{C}_{j,k}^{\text{split}}) = 1 - \alpha$. Here, $N^{\mathcal{D}_{\text{test}}}$ is the sample size of the test set $D_2 = (\boldsymbol{X}^{(2)}, \boldsymbol{Y}^{(2)})$.*

*Proof.* Recall our definition of $\widehat{\text{iLOCO}}_{j,k}(X_i^{(2)}, Y_i^{(2)})$ in Algorithm 1:

$$\widehat{\text{iLOCO}}_{j,k}(X_i^{(2)}, Y_i^{(2)}) = \hat{\Delta}_j(X_i^{(2)}, Y_i^{(2)}) + \hat{\Delta}_k(X_i^{(2)}, Y_i^{(2)}) - \hat{\Delta}_{j,k}(X_i^{(2)}, Y_i^{(2)}).$$

Since we have assumed that all samples of the test data $(X_i^{(2)}, Y_i^{(2)}) \overset{i.i.d.}{\sim} \mathcal{P}$, we note that $\mathbb{E}(\hat{\Delta}_j(X_i^{(2)}, Y_i^{(2)})|\boldsymbol{X}^{(1)}, \boldsymbol{Y}^{(1)}) = \Delta_j^{\text{split}}$, $\mathbb{E}(\hat{\Delta}_k(X_i^{(2)}, Y_i^{(2)})|\boldsymbol{X}^{(1)}, \boldsymbol{Y}^{(1)}) = \Delta_k^{\text{split}}$, $\mathbb{E}(\hat{\Delta}_{j,k}(X_i^{(2)}, Y_i^{(2)})|\boldsymbol{X}^{(1)}, \boldsymbol{Y}^{(1)}) = \Delta_{j,k}^{\text{split}}$, and hence

$$\mathbb{E}[\widehat{\text{iLOCO}}_{j,k}(X_i^{(2)}, Y_i^{(2)})|\boldsymbol{X}^{(1)}, \boldsymbol{Y}^{(1)}] = \Delta_j^{\text{split}} + \Delta_k^{\text{split}} - \Delta_{j,k}^{\text{split}} = \text{iLOCO}_{j,k}^{\text{split}}.$$

Furthermore, due to Assumption 8, the Lyapunov condition holds for $\widehat{\text{iLOCO}}_{j,k}(X_i^{(2)}, Y_i^{(2)}) - \mathbb{E}(\widehat{\text{iLOCO}}_{j,k}(X_i^{(2)}, Y_i^{(2)})|\boldsymbol{X}^{(1)}, \boldsymbol{Y}^{(1)})/\sigma_{j,k}^{\text{split}}$, conditional on the training set $\boldsymbol{X}^{(1)}, \boldsymbol{Y}^{(1)}$, and hence we can invoke the central limit theorem to obtain that

$$\frac{1}{\sigma_{j,k}^{\text{split}}\sqrt{N^{\mathcal{D}_{\text{test}}}}} \sum_{i=1}^{N^{\mathcal{D}_{\text{test}}}} [\widehat{\text{iLOCO}}_{j,k}(X_i^{(2)}, Y_i^{(2)}) - \text{iLOCO}_{j,k}^{\text{split}}] \overset{d}{\to} \mathcal{N}(0, 1).$$

Now, it remains to show the consistency of the variance estimate $\hat{\sigma}_{j,k} = \frac{1}{N^{\mathcal{D}_{\text{test}}} - 1} \sum_{i=1}^{N^{\mathcal{D}_{\text{test}}}} (\widehat{\text{iLOCO}}(X_i^{(2)}, Y_i^{(2)}) - \overline{\text{iLOCO}})^2$ for $(\sigma_{j,k}^{\text{split}})^2$.

Define the random variable $\xi_{N^{\mathcal{D}_{\text{test}}}, i} = [\widehat{\text{iLOCO}}_{j,k}(X_i^{(2)}, Y_i^{(2)}) - \text{iLOCO}_{j,k}^{\text{split}}]^2/(\sigma_{j,k}^{\text{split}})^2$. We aim to show that $\frac{1}{N^{\mathcal{D}_{\text{test}}} - 1} \sum_{i=1}^{N^{\mathcal{D}_{\text{test}}}} \xi_{N^{\mathcal{D}_{\text{test}}}, i} \overset{p}{\to} 1$. By Assumption 8, we have $\mathbb{E}|\xi_{N^{\mathcal{D}_{\text{test}}}, i}|^{3/2} = o(\sqrt{N})$. This implies the uniform integrability of $\{\xi_{N^{\mathcal{D}_{\text{test}}}, i}\}_i$:

$$\begin{aligned}
\mathbb{E}[|\xi_{N^{\mathcal{D}_{\text{test}}}, i}|\mathbb{I}(|\xi_{N^{\mathcal{D}_{\text{test}}}, i}| > x)] &\leq [\mathbb{E}|\xi_{N^{\mathcal{D}_{\text{test}}}, i}|^{3/2}]^{\frac{2}{3}}[\mathbb{P}(|\xi_{N^{\mathcal{D}_{\text{test}}}, i}| > x)]^{1/3} \\
&\leq [\mathbb{E}|\xi_{N^{\mathcal{D}_{\text{test}}}, i}|^{3/2}]^{\frac{2}{3}}[\frac{\mathbb{E}|\xi_{N^{\mathcal{D}_{\text{test}}}, i}|}{x}]^{1/3} \\
&= C^{2/3}x^{-1/3},
\end{aligned}$$

which converges to zero as $x \to \infty$. Here, we applied Holder's inequality on the first line, and applied Assumption 8 on the last line. With the uniform integrability of $\xi_{N^{\mathcal{D}_{\text{test}}}, i}$, we can follow the last part of the proof of Theorem 4 in Bayle et al. (2020), and show that

$$\frac{1}{N^{\mathcal{D}_{\text{test}}}} \sum_{i=1}^{N^{\mathcal{D}_{\text{test}}}} \xi_{N^{\mathcal{D}_{\text{test}}}, i} - 1 \overset{p}{\to} 0,$$

which then implies $\frac{1}{N^{\mathcal{D}_{\text{test}}}-1}\sum_{i=1}^{N^{\mathcal{D}_{\text{test}}}}\xi_{N^{\mathcal{D}_{\text{test}}},i}\xrightarrow{p}1$ as $N^{\mathcal{D}_{\text{test}}}\to\infty$, and hence $\hat{\sigma}_{j,k}/\sigma_{j,k}^{\text{MP}}\xrightarrow{p}1$.

Therefore, by Slutsky's theorem, we have

$$\frac{1}{\hat{\sigma}_{j,k}\sqrt{N^{\mathcal{D}_{\text{test}}}}}\sum_{i=1}^{N^{\mathcal{D}_{\text{test}}}}[\widehat{\text{iLOCO}}_{j,k}(X_i^{(2)},Y_i^{(2)})-\text{iLOCO}_{j,k}^{\text{split}}]\xrightarrow{d}\mathcal{N}(0,1),$$

which implies the asymptotically valid coverage of $\mathbb{C}_{j,k}^{\text{split}}$ for $\text{iLOCO}_{j,k}^{\text{split}}$. $\qquad\square$

## D  Inference Theory for iLOCO-MP

In this section, we restate the notations, assumptions and theorem in Section 3.2 of the main paper with more details.

**Inference target for iLOCO-MP**: Let

$$f(X)=\frac{1}{\binom{N}{n}\binom{M}{m}}\sum_{\substack{I\subset[N],|I|=n\\F\subset[M],|F|=m}}H(\boldsymbol{X}_{I,F},Y_F)(X^F)$$

be the minipatach ensemble predictor, when taking expectation over the randomly subsampled minipatches. The random minipatch ensemble $\hat{f}(X)=\frac{1}{B}\sum_{b=1}^{B}\hat{f}_b(X)$ converges to $f(X)$ as $B\to\infty$. Also define $f^{-j}(X)=\frac{1}{\binom{N}{n}\binom{M-1}{m}}\sum_{\substack{I\subset[N],|I|=n\\F\subset[M]\backslash j,|F|=m}}H(\boldsymbol{X}_{I,F},Y_F)(X^F)$, and $f^{-k},f^{-(j,k)}$ similarly. We can then write out the iLOCO inference target for minipatch ensembles as follows:

$$\text{iLOCO}_{j,k}^{\text{MP}}=\Delta_j^{\text{MP}}+\Delta_k^{\text{MP}}-\Delta_{j,k}^{\text{MP}},\tag{5}$$

where $\Delta_{j,k}^{\text{MP}}=\mathbb{E}_{X,Y}[\text{Err}(Y,f^{-(j,k)}(X^{-(j,k)};\boldsymbol{X}^{-(j,k)},\boldsymbol{Y}))-\text{Err}(Y,f(X;\boldsymbol{X},\boldsymbol{Y}))|\boldsymbol{X},\boldsymbol{Y}]$, and $\Delta_j^{\text{MP}},\Delta_k^{\text{MP}}$ are defined similarly.

For technical expositions, let

$$\begin{aligned}h_{j,k}(X,Y;\boldsymbol{X},\boldsymbol{Y})=&\text{Err}(Y,f^{-j}(X^{-j};\boldsymbol{X}^{-j},\boldsymbol{Y}))+\text{Err}(Y,f^{-k}(X^{-k};\boldsymbol{X}^{-k},\boldsymbol{Y}))\\&-\text{Err}(Y,f^{-(j,k)}(X^{-(j,k)};\boldsymbol{X}^{-(j,k)},\boldsymbol{Y}))-\text{Err}(Y,f(X;\boldsymbol{X},\boldsymbol{Y}))\end{aligned}$$

be the interaction importance score of feature pair $(j,k)$, when using the model trained on $(\boldsymbol{X},\boldsymbol{Y})$ to predict data $(X,Y)$. Our inference target $\text{iLOCO}_{j,k}^{\text{MP}}$ defined in equation 5 can also be written as follows:

$$\text{iLOCO}_{j,k}^{\text{MP}}=\mathbb{E}_{X,Y}[h_{j,k}(X,Y;\boldsymbol{X},\boldsymbol{Y})|\boldsymbol{X},\boldsymbol{Y}],$$

where $(X,Y)$ is independent of the training data $(\boldsymbol{X},\boldsymbol{Y})$.

**Function $h_{j,k}(X,Y)$ and variance $(\sigma_{j,k}^{\text{MP}})^2$**: Also define

$$h_{j,k}(X,Y)=\mathbb{E}_{\boldsymbol{X},\boldsymbol{Y}}[h_{j,k}(X,Y;\boldsymbol{X},\boldsymbol{Y})|X,Y],$$

where the expectation is taken over the training but not test data. Its variance $\text{Var}(h_{j,k}(X_i,Y_i))$ is denoted by $(\sigma_{j,k}^{\text{MP}})^2$.

Moreover, let $\hat{h}_{j,k}(X_i,Y_i;\boldsymbol{X}_{-i},\boldsymbol{Y}_{-i})=\widehat{\text{iLOCO}}_{j,k}(X_i,Y_i)$, the estimated pairwise interaction importance score at sample $i$ in Algorithm 2. Define $\tilde{h}_{j,k}(X_i,Y_i;\boldsymbol{X}_{-i},\boldsymbol{Y}_{-i})=h_{j,k}(X_i,Y_i;\boldsymbol{X}_{-i},\boldsymbol{Y}_{-i})-h_{j,k}(X_i,Y_i)$. Denote the trained predictor on each minipatch $(I,F)$ by $\hat{f}_{I,F}(\cdot)$.

**Assumption 9.** *The normalized interaction importance r.v. satisfies the third moment condition:* $\mathbb{E}[h_{j,k}(X,Y)-\mathbb{E}h_{j,k}(X,Y)]^3/(\sigma_{j,k}^{\text{MP}})^3\le C$.

**Assumption 10.** *The error function is Lipschitz continuous w.r.t. the prediction: for any $Y\in\mathbb{R}$ and any predictions $\hat{Y}_1,\hat{Y}_2\in\mathbb{R}^d$,*

$$|\text{Err}(Y,\hat{Y}_1)-\text{Err}(Y,\hat{Y}_2)|\le L\|\hat{Y}_1-\hat{Y}_2\|_2.$$

**Assumption 11.** *The prediction difference between different minipatches are bounded:* $\|\hat{f}_{I,F}(X) - \hat{f}_{I',F'}(X)\|_2 \leq D$ *for any $X$, any minipatches $I$, $I' \subset [N]$, $F$, $F' \subset [M]$.*

**Assumption 12.** *The minipatch sizes $(m, n)$ satisfy $\frac{m}{M}$, $\frac{n}{N} \leq \gamma$ for some constant $0 < \gamma < 1$, and $n = o\left(\frac{\sigma_{j,k}^{\mathrm{MP}}}{LD}\sqrt{N}\right)$.*

**Assumption 13.** *The number of minipatches satisfies $B \gg \left(\frac{D^2 L^2 N}{(\sigma_{j,k}^{\mathrm{MP}})^2} + 1\right)\log N$.*

**Theorem 4** (Coverage of iLOCO-MP). *Suppose the training data $(X_i, Y_i)$ are independent, identically distributed. Under Assumptions 9-13, we have $\lim_{N\to\infty} \mathbb{P}(\mathrm{iLOCO}_{j,k}^{\mathrm{MP}} \in \mathbb{C}_{j,k}^{\mathrm{MP}}) = 1 - \alpha$.*

*Proof.* Our proof closely follows the argument and results in Gan et al. (2022). First, we can decompose the estimation error of the iLOCO interaction importance score at sample $i$ as follows:

$$
\begin{aligned}
&\widehat{\mathrm{iLOCO}}_{j,k}(X_i, Y_i) - \mathrm{iLOCO}_{j,k}^{\mathrm{MP}} \\
&= h_{j,k}(X_i, Y_i) - \mathbb{E}[h_{j,k}(X_i, Y_i)] + \hat{h}_{j,k}(X_i, Y_i; \mathbf{X}_{-i}, \mathbf{Y}_{-i}) - h_{j,k}(X_i, Y_i; \mathbf{X}_{-i}, \mathbf{Y}_{-i}) \\
&\quad + \mathbb{E}[h_{j,k}(X_i, Y_i; \mathbf{X}_{-i}, \mathbf{Y}_{-i})|\mathbf{X}_{-i}, \mathbf{Y}_{-i}] - \mathrm{iLOCO}_{j,k}^{\mathrm{MP}} \\
&\quad + \tilde{h}_{j,k}(X_i, Y_i; \mathbf{X}_{-i}, \mathbf{Y}_{-i}) - \mathbb{E}[\tilde{h}_{j,k}(X_i, Y_i; \mathbf{X}_{-i}, \mathbf{Y}_{-i})|\mathbf{X}_{-i}, \mathbf{Y}_{-i}] \\
&=: h_{j,k}(X_i, Y_i) - \mathbb{E}[h_{j,k}(X_i, Y_i)] + \varepsilon_i^{(1)} + \varepsilon_i^{(2)} + \varepsilon_i^{(3)},
\end{aligned}
$$

where

$$
\begin{aligned}
\varepsilon_i^{(1)} &= \hat{h}_{j,k}(X_i, Y_i; \mathbf{X}_{-i}, \mathbf{Y}_{-i}) - h_{j,k}(X_i, Y_i; \mathbf{X}_{-i}, \mathbf{Y}_{-i}), \\
\varepsilon_i^{(2)} &= \mathbb{E}[h_{j,k}(X_i, Y_i; \mathbf{X}_{-i}, \mathbf{Y}_{-i}) \mid \mathbf{X}_{-i}, \mathbf{Y}_{-i}] - \mathrm{iLOCO}_{j,k}^{\mathrm{MP}}, \\
\varepsilon_i^{(3)} &= \tilde{h}_{j,k}(X_i, Y_i; \mathbf{X}_{-i}, \mathbf{Y}_{-i}) - \mathbb{E}[\tilde{h}_{j,k}(X_i, Y_i; \mathbf{X}_{-i}, \mathbf{Y}_{-i}) \mid \mathbf{X}_{-i}, \mathbf{Y}_{-i}].
\end{aligned}
$$

Let $\varepsilon^{(k)} = \frac{1}{\sigma_{j,k}^{\mathrm{MP}}\sqrt{N}}\sum_{i=1}^{N}\varepsilon_i^{(k)}, k = 1, \ldots, 3$, where $\sigma_{j,k}^{\mathrm{MP}}$ is the standard deviation of $h_{j,k}(X_i, Y_i)$, as we defined in notations. Assumption 9, $\{h_{j,k}(X_i, Y_i)\}_{i=1}^{N}$ satisfy the Lyapunov condition, and hence we can apply the central limit theorem to obtain that $\frac{1}{\sigma_{j,k}^{\mathrm{MP}}\sqrt{N}}\sum_{i=1}^{N} h_{j,k}(X_i, Y_i) - \mathbb{E}[h_{j,k}(X_i, Y_i)] \xrightarrow{d} \mathcal{N}(0,1)$. Following the same arguments as the proof of Theorem 1, 2 in Gan et al. (2022), we only need to show that $\varepsilon^{(1)}$, $\varepsilon^{(2)}$, $\varepsilon^{(3)}$ converge to zero in probability, and $\hat{\sigma}_{j,k} \xrightarrow{p} \sigma_{j,k}^{\mathrm{MP}}$, where $\hat{\sigma}_{j,k}^2 = \frac{1}{N-1}\sum_{i=1}^{N}(\widehat{\mathrm{iLOCO}}_{j,k}(X_i, Y_i) - \overline{\mathrm{iLOCO}}_{j,k})^2$ is defined as in Algorithm 2.

**Convergence of $\varepsilon^{(1)}$.** $\varepsilon^{(1)}$ characterizes the deviation of the random minipatch algorithm from its population version. In particular, by Assumption 10, we can write

$$
\begin{aligned}
|\varepsilon^{(1)}| \leq \frac{L}{\sigma_{j,k}^{\mathrm{MP}}\sqrt{N}}\sum_{i=1}^{N}\Big( &\|f_{-i}^{-(j,k)}(X_i) - \hat{f}_{-i}^{-(j,k)}(X_i)\|_2 + \|f_{-i}^{-j}(X_i) - \hat{f}_{-i}^{-j}(X_i)\|_2 \\
&+ \|f_{-i}^{-k}(X_i) - \hat{f}_{-i}^{-k}(X_i)\|_2 + \|f_{-i}(X_i) - \hat{f}_{-i}(X_i)\|_2\Big),
\end{aligned}
$$

where

$$
f_{-i}^{-(j,k)}(X_i) = \frac{1}{\binom{N-1}{n}\binom{M-2}{m}}\sum_{\substack{I\subset[N], |I|=n \\ F\subset[M], |F|=m}}\mathbb{I}(i \notin I)\mathbb{I}(j, k \notin F)\hat{f}_{I,F}(X_i),
$$

and

$$
\hat{f}_{-i}^{-(j,k)}(X_i) = \frac{1}{\sum_{b=1}^{B}\mathbb{I}(i \notin I_b)\mathbb{I}(j, k \notin F_b)}\sum_{b=1}^{B}\mathbb{I}(i \notin I_b)\mathbb{I}(j, k \notin F_b)\hat{f}_{I_b, F_b}(X_i).
$$

$f_{-i}^{-j}(X_i)$, $\hat{f}_{-i}^{-j}(X_i)$, $f_{-i}^{-k}(X_i)$, $\hat{f}_{-i}^{-k}(X_i)$, $f_{-i}(X_i)$, and $\hat{f}_{-i}(X_i)$ are defined similarly. We then follow the same arguments as those in Section A.7.1 of Gan et al. (2022) to bound the MP predictor differences. The only

difference lies in bounding $\|f_{-i}^{-(j,k)}(X_i) - \hat{f}_{-i}^{-(j,k)}(X_i)\|_2$, where we need to concentrate $\frac{\sum_{b=1}^{B} \mathbb{I}(i \in I_b)\mathbb{I}(j,k \notin F_b)}{B}$ around $\mathbb{P}(i \in I_b, j, k \notin F_b)$. Since Assumption 12 suggests that $\mathbb{P}(i \in I_b, j, k \notin F_b) \geq (1-\frac{n}{N})(1-\frac{m}{M})(1-\frac{m}{M-1})$ is lower bounded, all arguments in Gan et al. (2022) can be similarly applied here, which also use Assumption 11 and Assumption 13. These arguments then lead to $\lim_{N\to\infty} \mathbb{P}(|\varepsilon^{(1)}| > \epsilon) = 0$ for any $\epsilon > 0$.

**Convergence of $\varepsilon^{(2)}$.** $\varepsilon^{(2)}$ captures the difference in iLOCO importance scores caused by excluding one training sample. In particular, we can write

$$|\varepsilon^{(2)}| \leq \frac{L}{\sigma_{j,k}^{\mathrm{MP}}\sqrt{N}} \sum_{i=1}^{N} \Big( \mathbb{E}_X \|f_{-i}^{-(j,k)}(X) - f^{-(j,k)}(X)\|_2 + \mathbb{E}_X\|f_{-i}^{-j}(X) - f^{-j}(X)\|_2$$
$$+ \mathbb{E}_X\|f_{-i}^{-k}(X) - f^{-k}(X)\|_2 + \mathbb{E}_X\|f_{-i}(X) - f(X)\|_2 \Big),$$

where

$$f^{-(j,k)}(X) = \frac{1}{\binom{N}{n}\binom{M-2}{m}} \sum_{\substack{I \subset [N], |I|=n \\ F \subset [M], |F|=m}} \mathbb{I}(j,k \notin F)\hat{f}_{I,F}(X),$$

$$f^{-j}(X) = \frac{1}{\binom{N}{n}\binom{M-1}{m}} \sum_{\substack{I \subset [N], |I|=n \\ F \subset [M], |F|=m}} \mathbb{I}(j \notin F)\hat{f}_{I,F}(X),$$

$$f^{-k}(X) = \frac{1}{\binom{N}{n}\binom{M-1}{m}} \sum_{\substack{I \subset [N], |I|=n \\ F \subset [M], |F|=m}} \mathbb{I}(k \notin F)\hat{f}_{I,F}(X),$$

$$f(X) = \frac{1}{\binom{N}{n}\binom{M}{m}} \sum_{\substack{I \subset [N], |I|=n \\ F \subset [M], |F|=m}} \hat{f}_{I,F}(X),$$

and $f_{-i}^{*-(j,k)}(X)$, $f_{-i}^{*-j}(X)$, $f_{-i}^{*-k}(X)$, $f_{-i}^{*}(X)$ are defined as earlier. Following the same arguments as those in Section A.7.2 of Gan et al. (2022), we can bound the differences between the leave-one-out predictions and the full model predictions by $\frac{Dn}{N}$. Therefore, we have $|\varepsilon^{(2)}| \leq \frac{4LDn}{\sigma_{j,k}^{\mathrm{MP}}\sqrt{N}}$, and by Assumption 12, $\lim_{N\to\infty} \varepsilon^{(2)} = 0$.

**Convergence of $\varepsilon^{(3)}$.** Recall the definition of $\tilde{h}_j(X_i, Y_i; \mathbf{X}_{-i}, \mathbf{Y}_{-i})$, we can write

$$\varepsilon^{(3)} = h_{j,k}(X_i, Y_i; \mathbf{X}_{-i}, \mathbf{Y}_{-i}) - \mathbb{E}[h_{j,k}(X_i, Y_i; \mathbf{X}_{-i}, \mathbf{Y}_{-i}) \mid \mathbf{X}_{-i}, \mathbf{Y}_{-i}]$$
$$- h_{j,k}(X_i, Y_i) + \mathbb{E}[h_{j,k}(X_i, Y_i)].$$

The only difference of our proof here from Section A.7.3 in Gan et al. (2022) is that we are looking at the interaction score function $h_{j,k}$ instead of individual feature importance function $h_j$. Therefore, the main argument is to bound the stability notion in Bayle et al. (2020) associated with function $h_{j,k}$:

$$\gamma_{loss}(h_{j,k}) = \frac{1}{N-1} \sum_{l \neq i} \mathbb{E}\left[ \left( h'_{j,k}(X_i, Y_i; \mathbf{X}_{-i}, \mathbf{Y}_{-i}) - h'_{j,k}(X_i, Y_i; \mathbf{X}_{-i}^{\backslash l}, \mathbf{Y}_{-i}^{\backslash l}) \right)^2 \right],$$

where $(\mathbf{X}_{-i}, \mathbf{Y}_{-i})$ denotes the training data excluding sample $i$, while $(\mathbf{X}_{-i}^{\backslash l}, \mathbf{Y}_{-i}^{\backslash l})$ excludes sample $i$ and substitutes sample $l$ by a new sample $(X_{N+1}, Y_{N+1})$. The function $h'_{j,k}(X_i, Y_i; \mathbf{X}_{-i}, \mathbf{Y}_{-i}) = h_{j,k}(X_i, Y_i; \mathbf{X}_{-i}, \mathbf{Y}_{-i}) - \mathbb{E}[h_{j,k}(X_i, Y_i; \mathbf{X}_{-i}, \mathbf{Y}_{-i}) \mid \mathbf{X}_{-i}, \mathbf{Y}_{-i}]$. We note that

$$\gamma_{loss}(h_{j,k}) = \frac{1}{N-1} \sum_{l \neq i} \mathbb{E}\Bigg[ \mathrm{Var}\Big( h_{j,k}(X_i, Y_i; \mathbf{X}_{-i}, \mathbf{Y}_{-i})$$
$$- h_{j,k}(X_i, Y_i; \mathbf{X}_{-i}^{\backslash l}, \mathbf{Y}_{-i}^{\backslash l}) \Big| \mathbf{X}_{-i}, \mathbf{Y}_{-i}, X_{N+1}, Y_{N+1} \Big) \Bigg]$$
$$\leq \frac{1}{N-1} \sum_{l \neq i} \mathbb{E}\left[ \left( h_{j,k}(X_i, Y_i; \mathbf{X}_{-i}, \mathbf{Y}_{-i}) - h_{j,k}(X_i, Y_i; \mathbf{X}_{-i}^{\backslash l}, \mathbf{Y}_{-i}^{\backslash l}) \right)^2 \right].$$

Recall our definitions of $f_{-i}^{-j}(X)$, $f_{-i}^{-k}(X)$, $f_{-i}^{-(j,k)}(X)$ when showing the convergence of $\varepsilon^{(1)}$. In addition, we define $f_{-i}^{-j}(X; l \leftarrow N+1)$, $f_{-i}^{-k}(X; l \leftarrow N+1)$, $f_{-i}^{-(j,k)}(X; l \leftarrow N+1)$ as the corresponding predictors if the training sample $(X_l, Y_l)$ were substituted by a new sample $(X_{N+1}, Y_{N+1})$. Therefore, we can further show that

$$
\begin{aligned}
\gamma_{loss}(h_{j,k}) \leq &\frac{4L^2}{N-1} \sum_{l \neq i} \mathbb{E}\|f_{-i}^{-j}(X_i) - f_{-i}^{-j}(X_i; l \leftarrow N+1)\|_2^2 \\
&+ \mathbb{E}\|f_{-i}^{-k}(X_i) - f_{-i}^{-k}(X_i; l \leftarrow N+1)\|_2^2 \\
&+ \mathbb{E}\|f_{-i}^{-(j,k)}(X_i) - f_{-i}^{-(j,k)}(X_i; l \leftarrow N+1)\|_2^2 \\
&+ \mathbb{E}\|f_{-i}(X_i) - f_{-i}(X_i; l \leftarrow N+1)\|_2^2,
\end{aligned}
$$

where we have applied Assumption 10.

For any subset $I \subset [N]$ that includes sample $l$, let $I^{\backslash l} = \{i \neq l : i \in I\} \cup \{N+1\}$. We then have

$$
\begin{aligned}
&\|f_{-i}^{-(j,k)}(X_i) - f_{-i}^{-(j,k)}(X_i; l \leftarrow N+1)\|_2 \\
\leq &\frac{1}{\binom{N-1}{n}\binom{M-2}{m}} \sum_{\substack{I \subset [N], |I|=n \\ F \subset [M], |F|=m}} \mathbb{I}(i \notin I)\mathbb{I}(j,k \notin F)\mathbb{I}(l \in F)\|\hat{f}_{I,F}(X_i) - \hat{f}_{I^{\backslash l},F}(X_i)\|_2 \\
\leq &\frac{Dn}{N-1},
\end{aligned}
$$

where the last line is due to Assumption 11. Same bounds can also be shown for $\|f_{-i}^{-j}(X_i) - f_{-i}^{-j}(X_i; l \leftarrow N+1)\|_2$, $\|f_{-i}^{-k}(X_i) - f_{-i}^{-k}(X_i; l \leftarrow N+1)\|_2$, $\|f_{-i}(X_i) - f_{-i}(X_i; l \leftarrow N+1)\|_2$. Therefore, we have $\gamma_{loss}(h_{j,k}) \leq \frac{16L^2D^2n^2}{(N-1)^2}$. Applying Assumption 12, the rest of the arguments in Section A.7.3 of Gan et al. (2022) follow directly, and we have $\varepsilon^{(3)} \xrightarrow{p} 0$.

**Consistency of variance estimate.** The consistency of variance estimate $\hat{\sigma}_{j,k}^2$ to $(\sigma_{j,k}^{\mathrm{MP}})^2$ can also be shown following similar proofs of Theorem 2 in Gan et al. (2022). Similar to the proof of Theorem 3, the moment condition in Assumption 8 implies the uniform integrability of $h_{j,k}(X)/\sigma_{j,k}^{\mathrm{MP}}$, similar to the condition for the individual feature importance score in Gan et al. (2022). The main arguments are bounding the stability of the function $h_{j,k}$, and the difference between $h_{j,k}(X, Y; \mathbf{X}^{-i}, \mathbf{Y}^{-i})$ and $h_{j,k}(X, Y; \mathbf{X}, \mathbf{Y})$, which have already been shown earlier. Therefore, we can establish $\hat{\sigma}_{j,k}/\sigma_{j,k}^{\mathrm{MP}} \xrightarrow{p} 1$, which finishes the proof. $\qquad\square$

# E    A General Version of Proposition 1 and its Proofs

Suppose that $\mathbb{E}(Y|X) = f^*(X)$ for some unknown mean function $f^*(\cdot)$. The conditional distribution of $Y$ given $X$ will be specified later for regression and classification settings. Consider functional ANOVA decomposition for $f^*(X)$:

$$f^*(X) = g_0 + \sum_{j=1}^{M} g_j(X_j) + \sum_{1 \leq j < k \leq M} g_{j,k}(X_j, X_k) + \sum_{1 \leq j < k < l \leq M} g_{j,k}(X_j, X_k, X_l) + \cdots.$$

Following the notational convention of functional ANOVA, for any $u \subseteq [M]$, we let $g_u(X_u)$ denotes the function term with indices in $u$. We can then write $f^*(X) = \sum_{u \subseteq [M]} g_u(X_u)$, where $g_u(X_u) = g_0$ if $u = \emptyset$.

**Assumption 14.** *The functional ANOVA satisfies the following:*

- *Zero mean: $\mathbb{E}[g_u(X_u)] = 0$ when $u \neq \emptyset$.*

- *Zero correlation: $\mathbb{E}[g_u(X_u)g_v(X_v)] = 0$ if $u \neq v$.*

Note that much of the work on functional ANOVA assumes orthogonality of the functions and Assumption 13 can be viewed as a probabilistic extension of such conditions Hooker (2007). In fact, when all features are independent, Assumption 14 is immediately implied by defining $g_u(X_u)$ recursively: $g_0 = \mathbb{E}[f^*(X)]$, $g_u(X_u) = \mathbb{E}[f^*(X) - \sum_{v \subset u} g_v(X)|X_u]$ (see Proposition 3).

Consider the population S-way interaction iLOCO score: $\text{iLOCO}_S^* = \sum_{T \subseteq S, T \neq \emptyset} (-1)^{|T|+1} \Delta_T^*$, where $\Delta_T^* = \mathbb{E}(\text{Err}(Y, f^{*-T}(X_{-T})) - \mathbb{E}(\text{Err}(Y, f^*(X))$, with $f^{*-T}(X_{-T})$ being the population model excluding feature(s) $T$:

$$f^{*-T}(X_{-T}) = \sum_{u \subseteq [M] \setminus T} g_u(X_u).$$

When $S = \{j, k\}$, $\text{iLOCO}_S^* = \Delta_j^* - \Delta_k^* - \Delta_{j,k}^* = \text{iLOCO}_{j,k}^*$. As we will see in Proposition 3, in the independent feature setting, $f^{*-T}(X_{-T}) = \mathbb{E}(Y|X_{-T})$ is the oracle predictive function for $Y$ given $X_{-T}$.

The following proposition is a generalized version of Proposition 1 in the main paper.

**Proposition 2.** *Consider the functional ANOVA model under Assumption 14.*

(a) *Consider a regression model where $Y = f^*(X) + \epsilon$, with $\epsilon$ being zero-mean random noise with finite second moment, independent from $X$. If $\text{Err}(Y, \hat{Y}) = (Y - \hat{Y})^2$ is the mean squared error function, then*

$$\text{iLOCO}_S^* = \sum_{u \subseteq [M]: u \supseteq S} \mathbb{E}(g_u(X_u)^2). \tag{6}$$

(b) *Consider a classification model where $Y \sim Bernoulli(f^*(X))$. If $\text{Err}(Y, \hat{Y}) = |Y - \hat{Y}|$, then*

$$\text{iLOCO}_S^* = 2 \sum_{u \subseteq [M]: u \supseteq S} \mathbb{E}(g_u(X_u)^2).$$

Proposition 2 implies Proposition 1 in the main paper. It also justifies the higher-order S-way interaction iLOCO score defined in Section 2.4.

The following proposition provides an example where Assumption 14 easily holds.

**Proposition 3.** *Suppose that $\mathbb{E}(Y|X) = f^*(X)$ and features $X_1, \ldots, X_M$ are all independent. If $g_0 = \mathbb{E}(f^*(X))$, $g_u(X_u) = \mathbb{E}(f^*(X)|X_u) - \sum_{v \subset u} g_v(X_v)$ are defined recursively, then the functional ANOVA decomposition holds: $f^*(X) = \sum_{u \subseteq [M]} g_u(X)$, satisfying Assumption 14. Furthermore, $\mathbb{E}(Y|X_u) = \sum_{v \subseteq u} g_v(X_v)$.*

*Proof of Proposition 2.* We prove the results for regression and classification separately.

**Proof of (a).** We start from the regression model and the mean squared error loss. By the definition of $\Delta_T^*$, we can write

$$
\begin{aligned}
\Delta_T^* &= \mathbb{E}[(Y - f^{*-T}(X))^2] - \mathbb{E}[(Y - f^*(X))^2] \\
&= \mathbb{E}(\epsilon^2) + \mathbb{E}[(f^*(X) - f^{*-T}(X))^2] - \mathbb{E}(\epsilon^2) \\
&= \mathbb{E}[(\sum_{u \subseteq [M]: u \cap T \neq \emptyset} g_u(X_u))^2] \\
&= \sum_{u \subseteq [M]: u \cap T \neq \emptyset} \mathbb{E}[g_u^2(X_u)].
\end{aligned}
$$

where the second and last lines utilized the zero-mean and zero-correlation assumptions (Assumption 14) for our functional ANOVA decomposition.

Now we recall the definition of iLOCO$_S^*$ and plug in $\Delta_T^*$ above into iLOCO$_S^*$:

$$
\begin{aligned}
\text{iLOCO}_S^* &= \sum_{T \subseteq S, T \neq \emptyset} (-1)^{|T|+1} \Delta_T^* \\
&= \sum_{T \subseteq S, T \neq \emptyset} (-1)^{|T|+1} \sum_{u \subseteq [M]: u \cap T \neq \emptyset} \mathbb{E}[g_u^2(X_u)] \\
&= \sum_{u \subseteq [M]} \mathbb{E}[g_u^2(X_u)] \sum_{T \subseteq S, T \neq \emptyset} (-1)^{|T|+1} \mathbb{I}(u \cap T \neq \emptyset).
\end{aligned}
$$

In the following, we will show that

$$
\sum_{T \subseteq S, T \neq \emptyset} (-1)^{|T|+1} \mathbb{I}(u \cap T \neq \emptyset) = \mathbb{I}(u \supseteq S), \tag{7}
$$

which immediately implies equation 6.

To prove equation 7, we first let $A_j$ denote the event that $u \ni j$. we can then write

$$
\begin{aligned}
\sum_{T \subseteq S, T \neq \emptyset} (-1)^{|T|+1} \mathbb{I}(u \cap T \neq \emptyset) &= \sum_{T \subseteq S, T \neq \emptyset} (-1)^{|T|+1} \mathbb{I}(\cup_{j \in T} A_j), \\
\mathbb{I}(u \supseteq S) &= \mathbb{I}(\cap_{j \in S} A_j).
\end{aligned}
$$

The following lemma suggests $\mathbb{I}(\cap_{j \in S} A_j) = \sum_{T \subseteq S, T \neq \emptyset} (-1)^{|T|+1} \mathbb{I}(\cup_{j \in T} A_j)$ and hence concludes our proof for Part (a). Lemma 1 is essentially the inclusion-exclusion principle, but applied to indicator functions.

**Lemma 1.** *Consider arbitrary events $A_1, \ldots, A_n$ with $n \geq 2$. It holds that*

$$
\mathbb{I}(\cap_{j=1}^n A_j) = \sum_{T \subseteq [n], T \neq \emptyset} (-1)^{|T|+1} \mathbb{I}(\cup_{j \in T} A_j). \tag{8}
$$

**Proof of (b).** Now we consider the classification model $Y \sim \text{Bernoulli}(f^*(X))$ and the absolute error $\text{Err}(Y, \hat{Y}) = |Y - \hat{Y}|$. We first note that for any predictive model $\hat{Y} = f(X) \in [0, 1]$, we have

$$
\begin{aligned}
\mathbb{E}(\text{Err}(Y, f(X))) &= \mathbb{E}[(1 - f(X))\mathbb{P}(Y = 1) + f(X)\mathbb{P}(Y = 0)] \\
&= \mathbb{E}[(1 - f(X))f^*(X) + f(X)(1 - f^*(X))] \\
&= \mathbb{E}[f^*(X) + f(X) - 2f(X)f^*(X)].
\end{aligned} \tag{9}
$$

Therefore, we can write

$$
\Delta_T^* = \mathbb{E}[\text{Err}(Y, f^{*-T}(X_{-T})) - \text{Err}(Y, f^*(X))] = \mathbb{E}[(f^{*-T}(X_{-T}) - f^*(X))(1 - 2f^*(X))].
$$

Using the functional ANOVA decomposition, we can then write

$$
\begin{aligned}
\Delta_T^* &= \mathbb{E}[(f^{*-T}(X_{-T}) - f^*(X))(1 - 2f^*(X))] \\
&= \mathbb{E}[\sum_{u \subseteq [M]: u \cap T \neq \emptyset} g_u(X_u)(2f^*(X) - 1)] \\
&= 2\mathbb{E}[\sum_{u \subseteq [M]: u \cap T \neq \emptyset} g_u(X_u)f^*(X)] \\
&= 2\mathbb{E}[\sum_{u \subseteq [M]: u \cap T \neq \emptyset} g_u^2(X_u)],
\end{aligned}
$$

where the second line is due to the zero-mean assumption, and the last line is due to the zero-correlation assumption for our functional ANOVA decomposition.

Compared to the regression setting and mean squared error, $\Delta_T^*$ for classification and absolute error loss takes a very similar form, except with an extra factor 2. Therefore, using the same argument as those for proving Part (a), we can show that

$$
\text{iLOCO}_S^* = 2\mathbb{E}[\sum_{u \subseteq [M]: u \supseteq S} g_u^2(X_u)].
$$

The proof is now complete. $\qquad\square$

*Proof of Lemma 1.* We prove Lemma 1 by induction. When $n = 2$, we can immediately verify that $\mathbb{I}(\cap_{j=1}^n A_j) = \mathbb{I}(A_1 \cap A_2) = \mathbb{I}(A_1) + \mathbb{I}(A_2) - \mathbb{I}(A_1 \cup A_2)$. Now we assume that equation 8 holds for $n = k$, and then show that this implies equation 8 holds for $n = k + 1$. In particular, we can write

$$
\begin{aligned}
\mathbb{I}(\cap_{j=1}^{k+1} A_j) = \mathbb{I}((\cap_{j=1}^k A_j) \cap A_{k+1}) &= \mathbb{I}(\cap_{j=1}^k A_j) + \mathbb{I}(A_{k+1}) - \mathbb{I}((\cap_{j=1}^k A_j) \cup A_{j+1}) \\
&= \mathbb{I}(\cap_{j=1}^k A_j) + \mathbb{I}(A_{k+1}) - \mathbb{I}(\cap_{j=1}^k (A_j \cup A_{j+1})).
\end{aligned}
$$

Let $B_j = A_j \cup A_{j+1}$, and apply equation 8 on $\mathbb{I}(\cap_{j=1}^k A_j)$ and $\mathbb{I}(\cap_{j=1}^k B_j)$. We then have

$$
\begin{aligned}
\mathbb{I}(\cap_{j=1}^{k+1} A_j) &= \sum_{T \subseteq [k], T \neq \emptyset} (-1)^{|T|+1} \mathbb{I}(\cup_{j \in T} A_j) + \mathbb{I}(A_{k+1}) - \sum_{T \subseteq [k]} (-1)^{|T|+1} \mathbb{I}(\cup_{j \in T} B_j) \\
&= \sum_{T \subseteq [k], T \neq \emptyset} (-1)^{|T|+1} \mathbb{I}(\cup_{j \in T} A_j) + \mathbb{I}(A_{k+1}) + \sum_{T \subseteq [k], T \neq \emptyset} (-1)^{|T|+2} \mathbb{I}(\cup_{j \in T} A_j \cup A_{k+1}) \\
&= \sum_{T \subseteq [k], T \neq \emptyset} (-1)^{|T|+1} \mathbb{I}(\cup_{j \in T} A_j) + \mathbb{I}(A_{k+1}) + \sum_{T \subseteq [k], T \neq \emptyset} (-1)^{|T|+2} \mathbb{I}(\cup_{j \in T \cup \{k+1\}} A_j) \\
&= \sum_{T \subseteq [k]} (-1)^{|T|+1} \mathbb{I}(\cup_{j \in T} A_j) + \sum_{T \subseteq [k+1]: T \ni j} (-1)^{|T|+1} \mathbb{I}(\cup_{j \in T} A_j) \\
&= \sum_{T \subseteq [k+1]} (-1)^{|T|+1} \mathbb{I}(\cup_{j \in T} A_j).
\end{aligned}
$$

Therefore, equation 8 with $n = k$ implies equation 8 with $n = k + 1$. Since we already showed equation 8 holds for $n = 2$, by induction, our proof is complete. $\qquad\square$

*Proof of Proposition 3.* By the recursive definition of $g_u(X_u)$, we know that $g_{[M]}(X) = f^*(X) - \sum_{u \subset [M]} g_u(X_u)$, and hence the functional ANOVA decomposition $f^*(X) = \sum_{u \subseteq [M]} g_u(X_u)$ holds.

In the remaining proof, we will assume the following claim to be true. We will prove this claim in the end.

**Claim 1.** *For any non-empty set $u \subset [M]$ and its proper subset $v \subset u$, $\mathbb{E}[g_u(X_u)|X_v] = 0$.*

By letting $v = \emptyset$, Claim 1 immediately implies the zero-mean assumption in Assumption 14. Furthermore, for any index set $u \neq v \subseteq [M]$,

$$
\begin{aligned}
\mathbb{E}[g_u(X_u)g_v(X_v)] &= \mathbb{E}[\mathbb{E}(g_u(X_u)g_v(X_v)|X_{u\cap v})] \\
&= \mathbb{E}[\mathbb{E}(g_u(X_u)|X_{u\cap v})\mathbb{E}(g_v(X_v)|X_{u\cap v})] \\
&= 0,
\end{aligned}
$$

where the second line is due to the independence among all features; the last line utilizes Claim 1 and the fact that $u \cap v \subset u$, $u \cap v \subset v$.

Furthermore, for any feature set $u$, $\mathbb{E}(Y|X_u) = \mathbb{E}[\mathbb{E}(Y|X)|X_u] = \mathbb{E}[\sum_{v \subseteq [M]} g_v(X_v)|X_u] = \sum_{v \subseteq [M]} \mathbb{E}[g_v(X_v)|X_u]$. Due to the independence among features, when $v \neq u$, $\mathbb{E}[g_v(X_v)|X_u] = \mathbb{E}[g_v(X_v)|X_{u\cap v}]$. Claim 1 implies that $\mathbb{E}[g_v(X_v)|X_{u\cap v}] = 0$ whenever $u \cap v \neq v$, or equivalently, whenever $v \not\subseteq u$. Therefore, $\mathbb{E}(Y|X_u) = \mathbb{E}[\sum_{v \subseteq u} g_v(X_v)|X_u] = \sum_{v \subseteq u} g_v(X_v)$. Therefore, we have proved all statements in Proposition 3.

Now we prove Claim 1 via induction. We first note that when $u$ is of size 1, Claim 1 is implied by the zero-mean property of $g_j(X_j), j \in [M]$, which has been shown in the beginning of this proof. Now suppose that Claim 1 holds for $u$ of size $k$. Then if the size of $u$ is $k+1$, for any of its proper subset $v \subset u$, we can write

$$
\mathbb{E}[g_u(X_u)|X_v] = \mathbb{E}[f^*(X)|X_v] - \sum_{w \subset u} \mathbb{E}[g_w(X_w)|X_{v\cap w}].
$$

Since $w \subset u$ is of size at most $k$, we can apply Claim 1 on $g_w(X_w)$. When $w \not\subseteq v$, $v \cap w \subset u$, $\mathbb{E}[g_w(X_w)|X_{v\cap w}] = 0$. Hence we can further write

$$
\begin{aligned}
\mathbb{E}[g_u(X_u)|X_v] &= \mathbb{E}[f^*(X)|X_v] - \sum_{w \subseteq v} \mathbb{E}[g_w(X_w)|X_{v\cap w}] \\
&= \mathbb{E}[f^*(X)|X_v] - \sum_{w \subset v} g_w(X_w) - g_v(X_v) \\
&= 0,
\end{aligned}
$$

where the last line is due to the definition of $g_v(X_v)$. Therefore, Claim 1 holds for any non-empty set $u \subset [M]$. The proof is now complete. $\qquad\square$

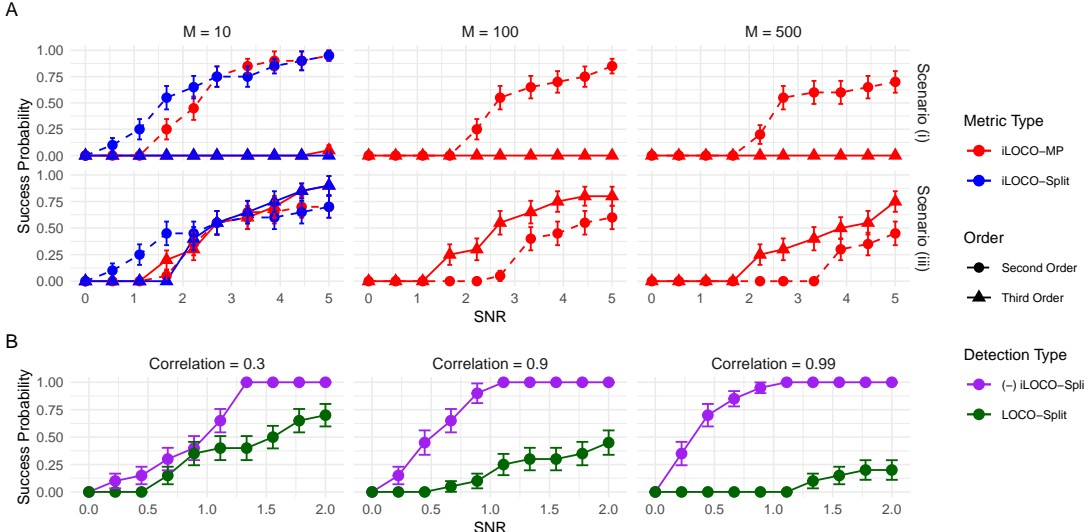

Figure A1: Part A shows the success probability of identifying an interaction pair in nonlinear classification scenarios (i) and (iii) using a RF classifier. Part B presents the success probability of detecting an important, correlated feature pair across varying correlation strengths.

# F    Additional Empirical Studies

We first include results on the same experiments as Figure 1 in the main paper, but for a Random Forest classifier as shown in Figure A1. Furthermore, we extend the experiments from Figure 2 in the main paper by including additional results on nonlinear regression, linear classification, and linear regression simulations, shown in Figures A2, A3, and A4, respectively. We also evaluate a correlated feature setting where we use an autoregressive design with $\mathbf{\Sigma}_{j,j+1}^{-1} = 0.8$ instead of $\mathbf{\Sigma} = \mathbf{I}$ in the uncorrelated simulations. Across all additional simulations, iLOCO-MP and iLOCO-Split consistently outperform baseline methods. Lastly, we include results validating our coverage guarantees using a setting with SNR = 2. Both iLOCO-MP and iLOCO-Split achieve empirical coverage near 0.9, supporting the validity of our inference procedure.

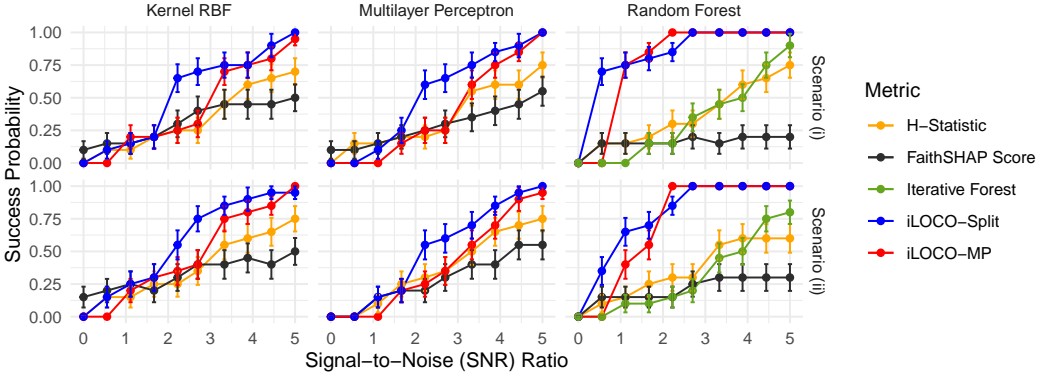

Figure A2: Ranking of feature pair (0,1) across SNR for KRBF, RF, and RF on nonlinear regression simulations 1 and 2.

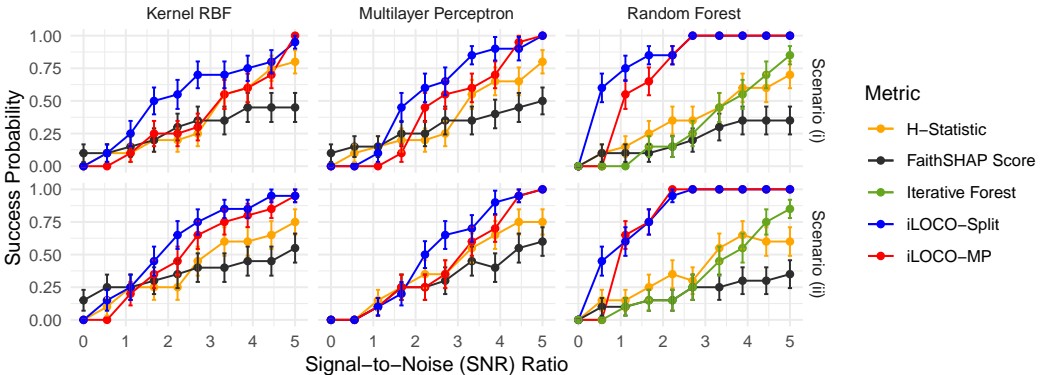

Figure A3: Ranking of feature pair (0,1) across SNR for KRBF, RF, and RF on linear classification simulations 1 and 2.

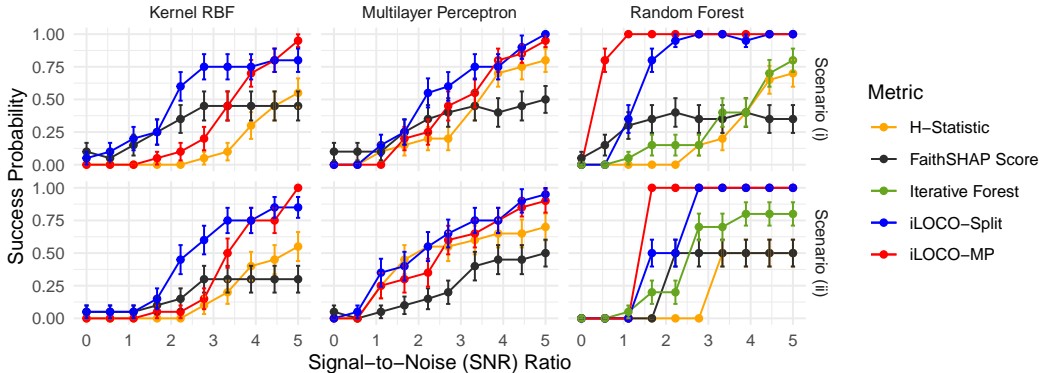

Figure A4: Ranking of feature pair (0,1) across SNR for KRBF, RF, and RF on linear regression simulations 1 and 2.

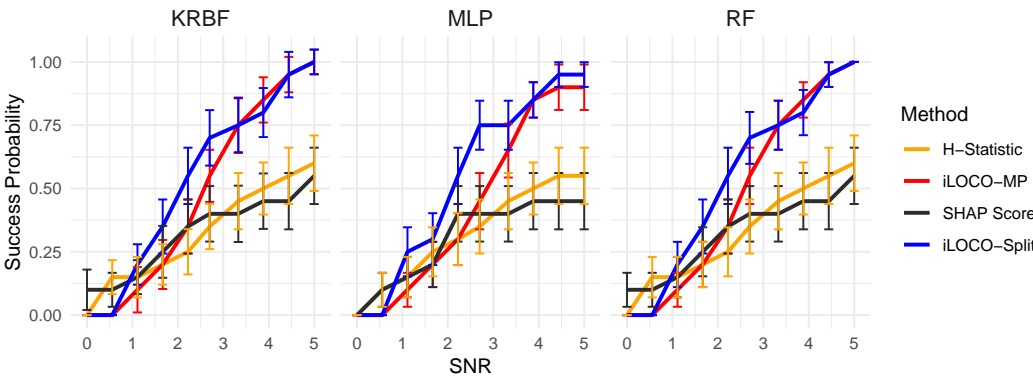

Figure A5: Ranking of feature pair (0,1) for KRBF, MLP, and RF regressors on the correlated simulation.

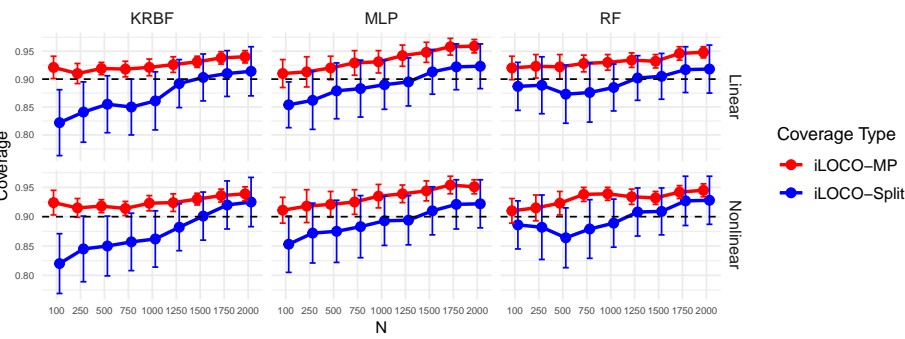

Figure A6: Coverage for the inference target of $snr = 2$ features of 90% confidence intervals in synthetic regression data using KRBF, MLP, and RF as the base estimators. iLOCO-MP and iLOCO-Split have valid coverage near 0.9.

