# OpenReview forum: "iLOCO: Distribution-Free Inference for Feature Interactions"
_TMLR — Decision pending for TMLR_

### Review · Reviewer_Muaw · 2026-04-20

**Summary Of Contributions:**

This paper proposes iLOCO, a model-agnostic metric for measuring feature interactions by extending the standard LOCO idea from single-feature importance to pairwise and higher-order interactions. It provides iLOCO via data-splitting and iLOCO via minipatch ensemble to estimate the iLOCO metric. The paper also provides asymptotic coverage results for both inference procedures and gives an ANOVA-style interpretation showing that iLOCO captures variance contributed by interaction terms under structured assumptions.

**Additional Comments:**

N/A

**Audience:**

Yes

**Audience Explanation:**

I think the paper would benefit some researchers in the TMLR community, particularly those working in interpretable machine learning, uncertainty quantification, and model-agnostic explanation methods. The problem of detecting and quantifying feature interaction, especially with statistical inference, is still relatively underdeveloped. The paper contributes both a new metric and inference procedures with theoretical guarantees and computational scalability.

**Claims And Evidence:**

Yes

**Claims Explanation:**

The claims in the paper are largely supported by accurate and clear evidence. I have checked the appendix. The appendix substantially strengthens the submission by providing detailed algorithms, formal inference targets, and complete proofs for the proposed confidence intervals. In particular, the iLOCO-Split estimator is supported by a well-justified asymptotic normality argument based on a conditional CLT and variance consistency, while the iLOCO-MP estimator is supported by a decomposition-based argument that extends prior minipatch inference results. The theoretical interpretation of iLOCO via functional ANOVA decomposition is also mathematically sound under the stated assumptions. Empirically, the paper provides extensive synthetic experiments that align with the theoretical claims, including interaction recovery and confidence interval coverage, as well as clear computational comparisons demonstrating scalability advantages. The evidence is presented clearly, with explicit algorithms, well-defined estimands, and interpretable figures.  Some claims are somewhat stronger than the supporting theory e.g., “distribution-free” inference, and this would benefit from more precise wording. The results are asymptotic and depend on iid sampling, bounded moments, stability conditions, and Lipschitz-type assumptions. So the paper does not establish truly assumption-free finite-sample inference. Overall, however, the main technical claims are supported by sufficiently rigorous and transparent evidence.

**Requested Changes:**

1. Tone down the phrase 'distribution-free inference': the current theory is asymptotic and assumption-based.I think'model-agnostic' or 'nonparametric' would be more appropriate.
2. The paper should explicitly state that the ANOVA-style interpretation of iLOCO depends on structured assumptions such as zero-mean / zero-correlation function components. This is a conditional interpretation, not a universal one.
3. The discussion of negative iLOCO values for important correlated pairs should be presented as an empirical observation or promising direction unless the authors add theory.
4. The inference targets for Split and MP are not identical, and this should be emphasized more clearly in the main paper.

---

> ### Author Response · Authors · 2026-06-22
>
> We appreciate the reviewer's careful read of our paper and thank them for their comments.
>
> *Tone down "distribution-free inference."*
>
> We thank the reviewer for this suggestion. We plan to revise the wording to more accurately reflect the nature of our theoretical guarantees. In particular, we will describe the inference as "assumption-light" and emphasize that the confidence intervals are asymptotically valid without making strong assumptions on the data-generating distribution or the employed learning algorithm, except for mild conditions such as boundedness. This will better reflect that our results are asymptotic and assumption-based, while still highlighting that they do not require smoothness, sparsity, or model-structure assumptions typical of many nonparametric inference procedures.
>
> *ANOVA-style interpretation relies on independent features.*
>
> We agree and plan to revise the presentation to make the conditions for the ANOVA-style interpretation explicit. Aside from presenting the zero-mean and zero-correlation assumptions on the functional ANOVA components before Proposition 1, we plan to add a paragraph after it to explain this more clearly. We will state that the clean variance-component interpretation of iLOCO holds when these assumptions hold and when the fitted models are replaced by their population counterparts. We will also note that feature correlations and estimation errors of the employed machine learning model can cause deviations from this clean representation, analogous to the known behavior of LOCO. Thus, the ANOVA interpretation will be presented as a structured, conditional interpretation rather than a universal characterization.
>
> *Negative iLOCO values for correlated, important features.*
>
> We thank the reviewer for this suggestion. We plan to revise the language around negative iLOCO values throughout the paper to better reflect that this is a promising empirical observation rather than a fully developed claim. In particular, we will describe this property as a potential additional usage of the iLOCO metric and as an observation worth noting for future investigation, rather than asserting that it solves the correlated feature problem.
>
> *Distinction between iLOCO-Split and iLOCO-MP inference targets.*
>
> We plan to add a clarification explaining that the iLOCO metric is defined relative to a particular collection of full and reduced models. Consequently, iLOCO-Split and iLOCO-MP estimate related but distinct targets. The data-splitting approach estimates the iLOCO metric associated with models trained by a chosen learning algorithm on the training split, and inference is conditional on that training split. In contrast, iLOCO-MP estimates the iLOCO metric associated with predictors that take the form of minipatch ensembles, built from any base learner and trained using the full data set. We will emphasize this distinction more clearly in the main text so that the inference targets for Split and MP are not conflated.

---

### Review · Reviewer_sE6X · 2026-05-06

**Summary Of Contributions:**

The paper proposes a model-agnostic framework for measuring feature interactions by extending the LOCO idea from single feature to higher-order interactions. This gives an intuitive inclusion-exclusion style interpretation of interaction strength. One important contribution is that the paper does not only define a score, but also develops asymptotically valid confidence intervals for interaction importance, together with a scalable minipatch version (iLOCO-MP) that reduces computational cost while still keeping good empirical performance. The theoretical analysis connects the population version of iLOCO to functional ANOVA decompositions, showing that under orthogonality-type assumptions the metric captures interaction-specific variance components. Empirically, the method performs well compared with baselines such as H-statistics and FaithSHAP on synthetic interaction recovery tasks.

Some claims could be stated more carefully. In particular, the “distribution-free” framing feels somewhat stronger than what is fully supported, since the inference results still rely on certain regularity assumptions. Additionally, the theoretical characterization is mainly for the population-level quantity, while the behavior for finite-sample is less directly analyzed. The empirical evaluation could be stronger with more real-world settings involving correlated features, more complex nuisance structure, and analysis of false-positive behavior.

**Audience:**

Yes

**Audience Explanation:**

The paper would be of interest to part of the TMLR audience due to the simplicity and interpretability of the proposed interaction metric, together with its practical applicability. The method combines an elegant theoretical formulation with scalable implementation and uncertainty quantification, making it potentially useful for researchers working on explainability, statistical machine learning, and feature interaction analysis in real applications.

**Claims And Evidence:**

Yes

**Claims Explanation:**

Yes. The main ideas in the paper are generally backed by both theory and experiments. The authors provide theoretical justification for the proposed interaction metric through its connection to functional ANOVA, and also develop an inference procedure with asymptotic guarantees.

On the empirical side, the method performs competitively in interaction recovery experiments and shows good scalability through the minipatch implementation.

**Requested Changes:**

1. $\textbf{Scalability}$  The higher-order extension is mathematically natural and elegant, but its computational burden grows rapidly with interaction order. The paper could discuss more explicitly what interaction orders are practically feasible in realistic settings, and whether the minipatch strategy is sufficient for scalable higher-order interaction search.


2. $\textbf{Distinction between the split and MP}$  Although iLOCO-Split and iLOCO-MP are both presented as approaches for estimating the interaction statistic, they appear to emphasize slightly different inferential targets and practical tradeoffs. Additional discussion clarifying their respective goals, assumptions, and interpretations would improve readability.

3. $\textbf{Practical interpretation and usage}$ The paper would benefit from more discussions on how practitioners should interpret significant interactions in real-world applications and how the proposed framework can guide downstream decision making. This would make the practical use cases more complete and accessible to a broader audience.

---

> ### Author Response · Authors · 2026-06-22
>
> We thank the reviewer for their feedback and are glad they found our method to be both scalable and broadly applicable.
>
> *Scalability of the higher-order extension.*
>
> We agree that higher-order iLOCO, while mathematically natural, introduces substantial computational challenges. We plan to add an explicit discussion in the higher-order interaction section. Under the data-splitting approach, estimating all nonempty subset contributions for an interaction set (S) requires fitting (2^|S| - 1) reduced models, so the cost grows exponentially with the interaction order. For the minipatch approach, although reduced models need not be explicitly refit, the number of minipatches may still need to grow exponentially in |S| under uniform sampling to ensure that all relevant subsets are adequately sampled. We will clarify that pairwise interactions are the primary practical target of the current method, while low-order extensions such as third-order interactions are feasible in moderate settings. We also plan to note that designing adaptive sampling strategies with minipatches could be a promising idea to further improve scalability for higher-order interaction search, and will leave this as an important direction for future work.
>
> *Distinction between iLOCO-Split and iLOCO-MP.*
>
> We plan to add a clarification explaining that the iLOCO metric is defined relative to a particular collection of full and reduced models. Consequently, iLOCO-Split and iLOCO-MP estimate related but distinct targets. The data-splitting approach estimates the iLOCO metric associated with models trained by a chosen learning algorithm on the training split, and inference is conditional on that training split. In contrast, iLOCO-MP estimates the iLOCO metric associated with predictors that take the form of minipatch ensembles, built from any base learner and trained using the full data set. We will also clarify the practical tradeoff: iLOCO-Split is conceptually straightforward and can be applied to arbitrary fitted models, whereas iLOCO-MP avoids refitting large models for each feature, uses the full dataset more efficiently, and provides a convenient, nearly cost-free post-training inference procedure for minipatch ensembles.
>
> *Practical interpretation and usage.*
>
> We thank the reviewer for this suggestion. We plan to add a new paragraph to the Discussion (Section 5) that connects our method back to the application domains introduced in Section 1, including drug discovery, genomics, and business analytics. This paragraph will clarify how significant iLOCO interactions and their confidence intervals can guide practical downstream decisions, such as prioritizing compounds for experimental validation or distinguishing true epistatic interactions from noise.

---

### Review · Reviewer_UMYo · 2026-06-09

**Summary Of Contributions:**

This paper proposes interaction LOCO (iLOCO), an extension of the feature importance index Leave-One-Covariate-Out (LOCO) to feature interactions. iLOCO measures the contribution of a pairwise interaction by comparing the change in prediction error when two features are removed individually and when they are removed simultaneously. The paper also extends the same idea to higher-order interactions involving three or more features.

Furthermore, this paper provides a procedure for constructing distribution-free confidence intervals for iLOCO and evaluating the uncertainty of interaction importance. In addition, to reduce the computational cost associated with LOCO-type methods, the authors introduce iLOCO-MP based on minipatch ensembles. Experiments verify the effectiveness of the proposed method in multiple simulation settings and on real data, comparing it with existing methods such as H-statistics, FaithSHAP, and Iterative Forest.


# Strengths
It defines interaction importance in a way that is consistent with the LOCO framework. While LOCO measures the change in prediction error when a single feature is removed, this paper compares the effect of removing two features separately and jointly, leading to an intuitive formulation of interaction importance. The paper also provides a theoretical interpretation of the quantity captured by iLOCO through a functional ANOVA decomposition, which gives theoretical support to the proposed index. The empirical evaluation is also relatively comprehensive, including comparisons with representative interaction detection methods, coverage validation for the proposed confidence intervals, and case studies on real datasets.


# Weaknesses
The core of the proposal is an extension of the existing LOCO index to interactions, and the methodological novelty appears relatively modest. In particular, iLOCO-MP and the confidence interval construction are closely related to existing LOCO inference and minipatch-ensemble-based inference procedures. Therefore, the novelty of the proposed method seems to lie not in the inference principle itself, but in applying the LOCO-type concept to the evaluation of feature interactions and organizing it into a practical estimation and inference procedure.

Another interesting aspect of the paper is the use of negative iLOCO values as a possible signal of correlated but important feature pairs. However, this point is treated mostly as a secondary observation in the current manuscript, and the empirical and conceptual support for this interpretation remains limited.

In addition, the presentation of the experimental settings and the readability of some figures could be improved.

**Audience:**

Yes

**Audience Explanation:**

Feature interactions are important in many applications, including scientific applications, medicine, and marketing, but they are more difficult to handle than individual feature importance. For example, several interaction indices based on Shapley-values have been proposed, but there remain multiple viewpoints on how interactions should be defined and axiomatized. In this context, the paper provides an intuitive LOCO-based definition and an inference procedure with confidence intervals, which could be a practical option for readers who want to evaluate the contribution of feature interactions to prediction performance.

**Broader Impact Concerns:**

I do not have specific broader impact concerns. The work is methodological, and I do not see ethical implications that require additional discussion beyond standard considerations for interpretability methods.

**Claims And Evidence:**

Yes

**Claims Explanation:**

The main claims of the paper are supported by accurate and convincing evidence. The formulation of iLOCO can be understood as a natural extension of LOCO, and the definitions for both pairwise and higher-order interactions are clear. The theoretical explanation based on functional ANOVA decomposition explains which interaction components the proposed index captures.

On the empirical side, the paper evaluates the method across multiple model classes, several simulation settings, and varying numbers of features and signal-to-noise ratios. It also includes comparisons with representative interaction detection methods, showing that the proposed method is competitive with existing methods and tends to perform stably, especially when the interaction signal is strong. The paper also validates the coverage of the proposed confidence intervals, which supports the validity of the proposed inference procedure.

However, some claims should be stated more carefully. In particular, the claim that uncertainty quantification for feature interactions has not been developed seems somewhat strong, and the relationship to prior work should be positioned more carefully.

**Requested Changes:**

# Critical changes for acceptance
The structure and explanation of the experiment section should be improved. In the current presentation, scenarios, the number of features, model classes, interaction orders, and correlation settings are introduced sequentially, which makes it difficult to follow what each experiment is intended to validate.

The meaning of the error bars in the figures should be clarified. The paper should state in the main text or figure captions whether they represent standard deviations, standard errors, or confidence intervals, and how many repetitions they are based on.

The claim that there is no prior work on uncertainty quantification for model-agnostic feature interactions seems somewhat strong, and the related work discussion should be expanded. For example, there are closely related lines of work, such as bootstrap confidence intervals for ALE-based interaction summaries and variance estimation for Shapley interaction estimators. The authors should clarify how these approaches differ from the LOCO-type, distribution-free confidence intervals proposed in this paper.

# Suggestions that would strengthen the work
The negative-iLOCO result in Figure 1(B), which suggests that negative iLOCO can detect correlated but important feature pairs, is interesting. However, given that Section 2.5 positions the correlated feature problem as an important challenge, the paper would be more convincing if this observation were evaluated more systematically. For example, the authors could consider settings where there is no true interaction but strongly correlated important features exist, settings where true interactions and correlation-induced redundancy coexist, and settings where correlated but unimportant feature pairs exist. Such experiments would make it clearer what negative iLOCO captures. In addition, unlike positive iLOCO, whose interpretation is supported by the functional ANOVA argument, the interpretation of negative iLOCO is mostly intuitive. A brief discussion of its relationship to redundancy, conditional feature importance, and related notions would help clarify the role of this observation.

---

> ### Author Response · Authors · 2026-06-22
>
> We thank the reviewer for their thoughtful feedback and are glad they found our contribution to be clear and intuitive.
>
> *Clarification on error bars.*
>
> We thank the reviewer for pointing this out. We plan to clarify in the simulation setup of Section 4.1 that error bars represent 95% confidence intervals constructed via the normal approximation described in Section 3: for each point, the interval is centered at the mean iLOCO estimate across samples, with width determined by the estimated standard deviation of the per-sample iLOCO scores scaled by the appropriate critical value. We will note that this differs from the coverage curves in Figure 3, where each point instead reflects empirical coverage computed by repeating the full estimation and interval-construction procedure across independent replicates.
>
> *Structure and explanation of the experiment section.*
>
> We agree that the experiment section would benefit from a clearer explanation of what each experiment validates. We plan to add an overview at the beginning of Section 4.1 that previews the three goals of our empirical studies: validating that iLOCO recovers true interactions, comparing iLOCO against existing detection methods, and confirming valid coverage of our confidence intervals. We will also add a brief orienting sentence at the start of each subsequent paragraph to clarify what each experiment is intended to validate before presenting the setup and results.
>
> *Prior work on uncertainty quantification for model-agnostic feature interactions.*
>
> We thank the reviewer for pointing this out. We plan to revise the related work discussion (Section 1.1) to better acknowledge existing uncertainty summaries for model-agnostic interaction methods, including bootstrap confidence intervals for Accumulated Local Effects (ALE) and variance estimates for Shapley-value-based interaction estimators. We will clarify that ALE-based procedures target curve- or surface-level summaries of a fixed trained model and are mainly useful for visualization, rather than predictive-risk-based global interaction measures designed for interaction detection. We will also clarify that existing bootstrap intervals for ALE are primarily practical uncertainty summaries rather than a general inference framework with formal validity guarantees. In particular, since ALE is estimated on a finite grid, often with data-dependent bins, classical bootstrap theory may apply to the discretized estimator with fixed bins but would not by itself imply valid inference for the original integral-form ALE functional. For Shapley-value-based interactions, we will explain that existing variance estimates often quantify Monte Carlo approximation error from sampling coalitions in the combinatorial Shapley formula, which is distinct from finite-sample statistical uncertainty. We also plan to tone down phrasing such as "first inference method for model-agnostic feature interactions" throughout the paper, and instead write that our approach provides, to the best of our knowledge, the first inferential framework with formal validity guarantees for detecting model-agnostic feature interactions.
>
> *Negative iLOCO values for correlated, important features.*
>
> We thank the reviewer for this suggestion. We plan to revise the language around negative iLOCO values throughout the paper to better reflect that this is a promising empirical observation rather than a fully developed claim. In particular, we will describe this property as a potential additional usage of the iLOCO metric and as an observation worth noting for future investigation, rather than asserting that it solves the correlated feature problem.
>
> *Practical interpretation and real-world usage.*
>
> We thank the reviewer for this suggestion. We plan to add a new paragraph to the Discussion (Section 5) that connects our method back to the application domains introduced in Section 1, including drug discovery, genomics, and business analytics. This paragraph will clarify how significant iLOCO interactions and their confidence intervals can guide practical downstream decisions, such as prioritizing compounds for experimental validation or distinguishing true epistatic interactions from noise.